# Observational Scaling Laws and the Predictability of Language Model Performance

**Yangjun Ruan**[1,2,3]
yjruan@cs.toronto.edu

**Chris J. Maddison**[2,3]
cmaddis@cs.toronto.edu

**Tatsunori Hashimoto**[1]
thashim@stanford.edu

[1]Stanford University    [2]University of Toronto    [3]Vector Institute

## Abstract

Understanding how language model performance varies with scale is critical to benchmark and algorithm development. Scaling laws are one approach to building this understanding, but the requirement of training models across many different scales has limited their use. We propose an alternative, *observational* approach that bypasses model training and instead builds scaling laws from ∼100 publically available models. Building a single scaling law from multiple model families is challenging due to large variations in their training compute efficiencies and capabilities. However, we show that these variations are consistent with a simple, generalized scaling law where language model performance is a function of a low-dimensional capability space, and model families only vary in their efficiency in converting training compute to capabilities. Using this approach, we show the surprising predictability of complex scaling phenomena: we show that several emergent phenomena follow a smooth, sigmoidal behavior and are predictable from small models; we show that the agent performance of models such as GPT-4 can be precisely predicted from simpler non-agentic benchmarks; and we show how to predict the impact of post-training interventions like Chain-of-Thought and Self-Consistency as language model capabilities continue to improve.

## 1   Introduction

Language model (LM) scaling plays a central role in discussions of model capabilities and affects everything from the tasks they can perform to the effectiveness of post-training techniques such as Chain-of-Thought [99]. Due to this importance, understanding and predicting LM behaviors across scales, benchmarks, and algorithmic interventions is a major question for many researchers and engineers. Machine learning researchers may wish to understand whether their proposed algorithmic interventions remain effective in the face of future model scaling, while engineers and benchmark builders may wish to understand whether complex capabilities such as agentic abilities will scale predictably in the same way as existing LM benchmarks.

Scaling laws [6, 36, 37, 44, 65] have been powerful tools for understanding the scaling trend of LMs, which have shown that LMs follow a precise power-law relationship between compute measures (such as training FLOPs) and downstream capabilities ranging from perplexity [37, 44] to benchmark performance [34, 35]. This power-law relationship has been used in a variety of ways – including hyperparameter and architecture selection [9, 37, 44] as well as model capability forecasting [25, 66, 67]. Unfortunately, scaling analyses remain uncommon in many benchmarking and post-training studies, as most researchers do not have the compute resources to build scaling laws from scratch, and open models are trained at too few scales (3-5) for reliable scaling predictions.

We show that many scaling analyses, such as understanding complex LM capabilities (e.g., "emergent" behaviors) and post-training interventions, can be done with a lower-cost, higher-resolution, and broader-coverage alternative to the standard approach of training LMs across compute scales.

The starting point of our work is the observation that there now exist hundreds of open models spanning a large range of scales and capabilities. While we cannot directly use these models for

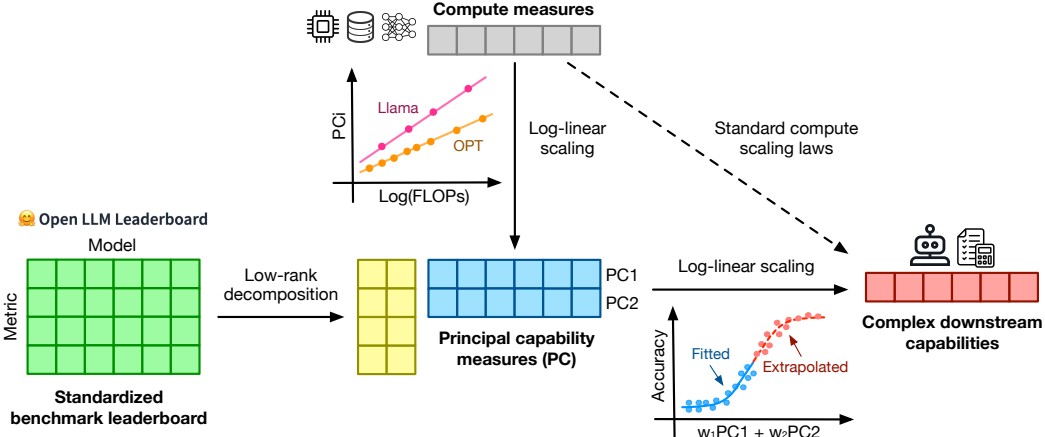

Figure 1: Observational scaling laws generalize existing compute scaling laws which directly relate training compute to downstream capabilities (dashed line) by hypothesizing the existence of a low-rank space of LM capabilities that have a log-linear relationship with compute (center), and can be extracted directly from standardized LM benchmarks (left). This enables us to get low-cost, high-resolution scaling predictions of LMs' complex downstream capabilities from their observable standard benchmark metrics using nearly 100 publicly accessible LMs (left to right).

compute scaling laws (as the training compute efficiency varies widely across model families), we might hope that there exists a more general scaling law that holds across model families. In particular, we hypothesize that the downstream performance of an LM is a function of a low-dimensional space of capabilities (e.g., natural language understanding, reasoning, and code generation), and that model families vary only in the efficiency by which they convert training compute to these capabilities. If such a relationship held, it would imply that there is a log-linear relationship from low-dimensional capabilities to downstream capabilities *across* model families (which would allow us to build scaling laws that leverage all existing models), as well as a log-linear relationship between training compute and capabilities *within* each model family (as in standard compute scaling) (Fig. 1).

Through an analysis of standard LM benchmarks (e.g., Open LLM Leaderboard [8]), we find a few such capability measures that have scaling relationships with compute within model families ($R^2 > 0.9$) (Fig. 3), and with downstream metrics across families. We call such relationships *observational* scaling laws as they predict complex downstream capabilities from simple observable quantities that we expect to scale with compute (like standardized benchmark performance)

The ability to build scaling laws across a large number of existing LMs from their standard benchmark metrics has significant advantages in cost, resolution, and coverage: Observational scaling incurs no training cost, while leveraging models spanning a much larger compute range than any single model family. It also significantly increases the resolution of scaling laws by virtue of using more models, which is useful for studying nearly discontinuous phenomena like "emergent" capabilities. Finally, observational scaling can combine model families from heterogeneous sources with very different scaling properties (e.g., LLaMA [91] vs StarCoder [48]) which allows us to study how different scaling strategies impact downstream performance and algorithmic interventions.

Finally, we show that using observational scaling laws is low-cost and straightforward, as there are a few model families that are sufficiently representative to replicate many of our core findings (Sec. 5). By using these representative families, we find that future works can easily make scaling predictions on benchmarks and post-training interventions by evaluating only 10-20 models.

We demonstrate the utility of observational scaling laws in three different settings that are challenging for compute scaling laws but are accurately predicted by ours: (i) **Emergent capabilities** (Sec. 4.1): We show that the high resolution of observational scaling laws reveals that the emergent behaviors of LMs [98] follow a smooth sigmoid, and can be predicted accurately using sub Llama-2 7B models. (ii) **Agentic capabilities** (Sec. 4.2): We show that the more complex capabilities of LMs as agents, as measured by AgentBench [57] and AgentBoard [61], can be predicted with simple benchmark metrics. Our scaling law precisely predicts the GPT-4 performance using weaker models (sub GPT-3.5) and identifies programming capabilities as driving agent performance. (iii) **Post-training interventions** (Sec. 4.3): We show that our scaling laws can reliably predict the gains of post-training techniques, such as CoT [99] and Self-Consistency [97] at scale, even when they are fitted on weak models (sub

Llama-2 7B). Finally, we show how to select only 10-20 representative models to replicate our core findings, making our scaling analyses more accessible with a low cost (Sec. 5).

## 2 Related Work

In this section, we briefly review the most relevant related work on downstream scaling laws and benchmark correlations. We include an extended related work discussion in Appx. C.

**Downstream scaling laws** Scaling laws have been generalized beyond pretraining loss to analyze transfer learning [1, 35, 85] and downstream performance [15, 30, 34] across various domains. However, whether the LM downstream performance demonstrates a rapid "emergence" or is predictable with scaling laws remains debated [23, 27, 38, 39, 60, 79, 83, 98, 101]. Finnveden [25] and Owen [67] have investigated the use of linear and sigmoidal scaling laws, derived from pretraining loss or computational measures, to extrapolate the benchmark performance. Arora and Goyal [5] derived a theory characterizing how LMs' complex skills can be derived as a composition of base skills. Our work differs in that we build practical higher-resolution scaling laws to predict LM downstream performance using multiple model families and their observable standard benchmark metrics.

**Correlations between benchmarks** Numerous works have studied the correlations between NLP benchmarks in vairous contexts [56, 68, 69, 71]. Most relevant to our work, Ilić [40] found that a single factor explains 85% of the variation on the Open LLM Leaderboard [8] and GLUE leaderboard [95], while Burnell et al. [14] extracted three factors for LM capabilities that account for 82% of the variation on HELM [51], aligning with our observations. Our work also observes such benchmark correlations and low-rank structures but is unique in utilizing these properties for the purpose of scaling predictions that can be used directly for benchmark and algorithm development.

## 3 Observational Scaling Laws

In this section, we introduce our observational scaling laws that generalize the standard compute scaling laws (Sec. 3.1). The key idea is to extract a low-dimensional capability measure for LMs from their observable benchmark performance (Sec. 3.2), which we find has a log-linear relationship with compute scale measures (Sec. 3.3) and can thus be used as surrogate "scale" for scaling analysis of complex LM capabilities (Sec. 3.4).

### 3.1 Generalizing Compute Scaling Laws

**Standard compute scaling** In *compute* scaling laws, there is a hypothesized power-law relationship between models' compute measures $C_m$ (e.g., training FLOPs) and their errors $E_m$ (e.g., perplexity). Specifically, for a model $m$ within a family $f$ (e.g., Llama-2 7B, 13B, and 70B) we hypothesize

$$\log(E_m) \approx \beta_f \log(C_m) + \alpha_f, \tag{1}$$

and if this linear fit is sufficiently accurate, we draw inferences about the performance of a model at future compute scales $C' > C$ by extrapolating this relationship. However, fitting such a scaling law can be tricky, as each model family $f$ and downstream benchmark has its own scaling coefficients $\beta_f$ and $\alpha_f$. This means that scaling experiments, especially for post-training analysis, are often fitted on very few (3-5) models sharing the same model family, and any predictions are valid only for a specific scaling strategy used within a model family.

Several studies [e.g., 25, 67] have generalized the functional form to analyze the scaling of LMs' downstream performance (where $E_m$ is normalized to $[0, 1]$) with a sigmoidal link function $\sigma$:

$$\sigma^{-1}(E_m) \approx \beta_f \log(C_m) + \alpha_f, \tag{2}$$

**Observational scaling** In our work, we hypothesize the existence of a low-dimensional capability measure for LMs that relate compute to more complex LM capabilities and can be extracted from observable standard LM benchmarks, as illustrated in Fig. 1. Specifically, given $T$ simple benchmarks and $B_{i,m}$ the error of a model $m$ on benchmark $i \in [T]$, we hypothesize that there exists some *capability vector* $S_m \in \mathbb{R}^K$ such that,

$$\sigma^{-1}(E_m) \approx \beta^\top S_m + \alpha \tag{3}$$

$$S_m \approx \theta_f \log(C_m) + \nu_f \tag{4}$$

$$B_{i,m} \approx \gamma_i^\top S_m. \tag{5}$$

for $\theta_f, \nu_f, \beta \in \mathbb{R}^K$, $\alpha \in \mathbb{R}$, and orthonormal vectors $\gamma_i \in \mathbb{R}^K$.

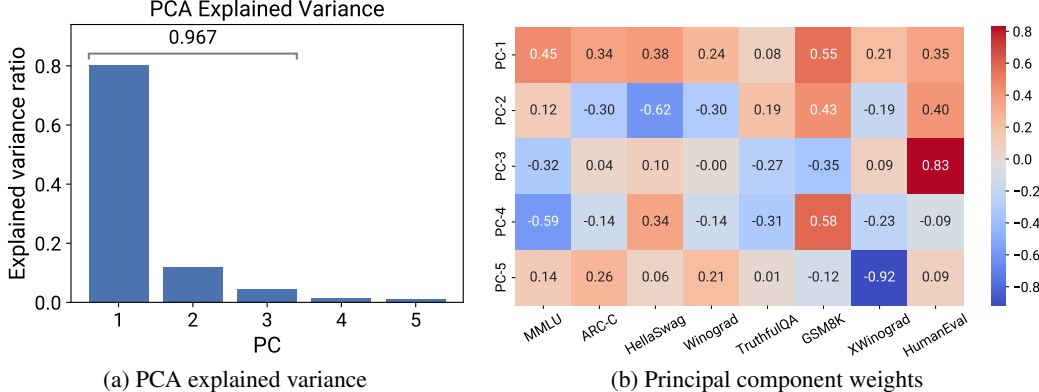

(a) PCA explained variance  (b) Principal component weights

Figure 2: Just a few capability dimensions explain most variability on a diverse range of standard LM benchmarks. We find that (a) the benchmark-model matrix is **low-dimensional** with the top 3 PCs explaining $\sim 97\%$ of the variance and (b) the PCs are **interpretable**: PC-1, PC-2, and PC-3 emphasize LMs' general, reasoning, programming capabilities, respectively.

We can view Eq. (3) and Eq. (4) as a generalization of Eq. (2), since combining them can recover the original scaling relationships for a single model family. However, when there are multiple model families, $S_m$ serves as a shared, low-dimensional space of model capabilities from which all downstream metrics ($E$ and $B$) are derived (as indicated by the absence of $f$ in Eq. (3) and Eq. (5)), and model families only vary in their efficiency in converting compute into capabilities (Eq. (4)). One useful way of interpreting Eq. (4) is that $\theta_f$ represents the compute efficiency of a model family $f$, and $S_m$ is the capabilities of model $m$ expressed in terms of log-FLOPs for this model family.

Finally, Eq. (5) ensures that these capabilities are not latent variables to be estimated for each model family, but are instead functions of fully observable properties ($B$). Since $\gamma \in \mathbb{R}^{K \times T}$ is orthonormal, we can linearly estimate $\hat{S}_m := \gamma B_m$, which makes our scaling analysis significantly more robust. Importantly, this enables us to apply this to a large number of public models from heterogeneous sources, including those proprietary ones without any public information on $C$ such as GPT-4.

### 3.2 Identifying a Low-Dimensional Capability Space (Eq. (5))

We validate the existence of a low-dimensional capability measure $S$ that linearly relates to standard LM benchmarks $B$ by showing that only a few principal components of $B$ capture most of its variation (Eq. (5)). We demonstrate that the benchmark-model matrix $B$ for a reasonable, broad set of benchmarks and models is low-rank and that Eq. (5) is a reasonable assumption.

**Models** Since the benchmark-model matrix $B$ can be directly measured for any LM, we include a large number of publicly accessible models for subsequent analysis. We collected a broad set of open LMs covering 21 model families (a collection of models across scales such as LLaMA-2 7B, 13B, 70B) and a total of 77 models. These encompass models trained from heterogeneous recipes, including standard training recipes like LLaMA [91], those trained on synthetic data like Phi [50], and models specifically trained on code data like StarCoder [48]. For this analysis, we consider only pretrained base models to avoid the complexities introduced by instruction tuning. We also include an analysis for instruction-tuned models that include proprietary ones like GPT-4 [66] in Appx. E.1, which demonstrates similar results. See Table D.1 for a detailed list of collected models.

**Benchmarks** We collected a set of diverse benchmarks that assess various LMs' capabilities. These include popular aggregated benchmarks like MMLU [32] that assess the general knowledge of LMs. For more specialized evaluations, we included ARC-C [19], HellaSwag [108], Winogrande [77] for commonsense reasoning, GSM8K [20] for mathematical reasoning, HumanEval [16] for programming, TruthfulQA [53] for truthfulness, and XWinograd [64] for multilingual capabilities. We carefully collected these metrics from standardized evaluation protocols for comparability across LMs. In particular, we compiled them from standardized leaderboards, like the Open LLM Leaderboard [8] and EvalPlus [55], when available. Otherwise, we used standardized libraries such as the LM Eval Harness [28] to evaluate the LMs. See Appx. D.1 for full details of our data collection pipeline.

**PCA analysis** After obtaining the benchmark metrics for the LMs, we addressed potential missing values (less than $1\%$ of all data), which may have occurred due to evaluation failures, by using PCA imputation. Subsequently, we applied PCA to extract the principal components of the evaluation metrics as the "principal capability" (PC) measures $S$ (additional details in Appx. D.3).

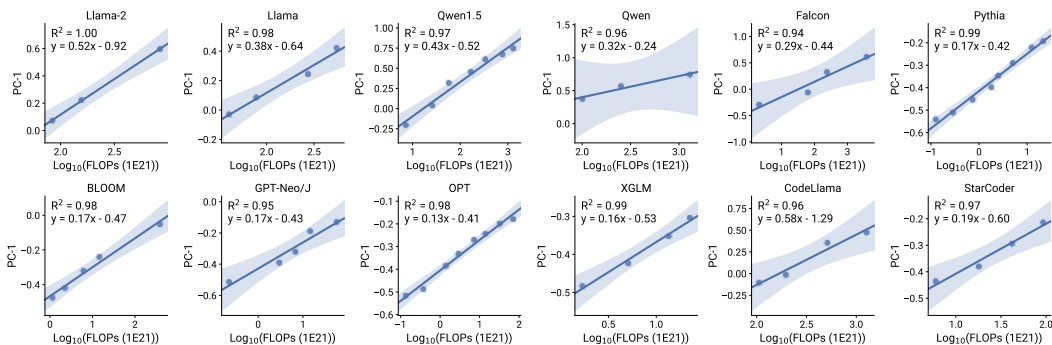

Figure 3: The extracted PC measures *linearly correlate* with log-compute within each model family. The linearity generally holds for various model families, and also for lower-ranked PCs (Fig. E.2).

**PC measures are low-dimensional**  We observe that the extracted PC measures are predominantly low-rank, with the top 3 PCs explaining $\sim 97\%$ of the variance, which supports a low-dimensional representation of benchmarks $B$ (Fig. 2a). Surprisingly, we find that the first PC alone explains nearly 80% of the variation in LM capabilities. Taking a closer look at these PCs, we find that these capability measures represent interpretable directions in which LMs capabilities may naturally vary as a function of scale (Fig. 2b). Specifically, PC-1 represents the "general capability" as a weighted average of all metrics; PC-2 corresponds to the "reasoning capability", emphasizing mathematical and coding benchmarks; and PC-3 primarily reflects the "programming capability". These findings suggest that many simple LM capabilities (as covered in our benchmarks) can be expressed as a linear combination of just a few "principal capabilities" $S$.

### 3.3 Principal Capability Measures as Surrogate Scale Measures (Eq. (4))

We now show that the PC measures $S$ scale log-linearly with training FLOPs within each model family, and can thus be interpreted as a cross-family generalization of compute $C$. We discuss some additional applications of PC measures as a smooth cross-family evaluation metric in Appx. B.

**Setup**  We collected all available information about training FLOPs on each of our models, analyzing papers and other public information to identify model size $N$ and pretraining data size $D$. For the models where we were able to identify this information, we used the simple estimate of $C \approx 6ND$ to obtain model training FLOPs [44]. See Table D.1 for our collected compute measures.

**PC measures linearly correlate with log-compute measures**  Fig. 3 illustrates the correlation between the top PC-1 measure with the corresponding training FLOPs for models within each model family. We find that for each model family with controlled training recipes and comparable compute scale measures, the LMs' PC-1 measure *linearly* correlates with their log-training FLOPs (with $R^2 > 0.9$). This linear correlation holds across a broad range of model families including those specifically trained on multilingual data like BLOOM [100] or those on code like StarCoder [48]. It also generally holds for lower-ranked PCs such as PC-2 and PC-3, as shown in Fig. E.2. Together with Sec. 3.2, these results support the validity of Equations (4) and (5), in which we hypothesized that models share the same capability space and a log-linear relationship determines the efficiency by which each model family converts their compute into these principal capabilities.

### 3.4 Fitting Observational Scaling Laws (Eq. (3))

**Fitting regression with PC measures**  Given a certain downstream error metric $E$ normalized to $[0, 1]$ that measures certain LM capabilities, we slightly generalize Eq. (3) to

$$E_m \approx h\sigma(\beta^\top S_m + \alpha) \qquad (6)$$

where $\beta \in \mathbb{R}^K$ and $\alpha \in \mathbb{R}$ are the regression weights and bias, $h \in [0, 1]$ is the sigmoidal scale that accounts for the potential discrepancies in the floor performance. We fit the regression with ordinary least squares and restrict $h \in [0.8, 1.0]$, which results in $h^* = 1$ in most experiments.

**Defining interpretable compute-like measures**  Recall that the core component of our scaling law is the fitted linear transformation $P_m := \beta^{*\top} S_m + \alpha^*$ that maps the extracted PCs into a scalar capability measure for a target downstream metric. While this is perfectly acceptable for prediction, our scaling analysis would be more interpretable if we expressed capabilities in units of FLOPs rather than an arbitrary scalar measure. We can achieve this by utilizing the fact that for a single family $f$, our observational scaling law reduces to a compute scaling law (Eq. (3) & Eq. (4)). Specifically, we

note that when Eq. (4) holds exactly, we have that for a model $m$ within a family $f$,

$$P_m := \beta^{*\top} S_m + \alpha^* = w_f \log(C_m) + b_f \tag{7}$$

where $w_f = \beta^{*\top} \theta_f$ and $b_f = \beta^{*\top} \nu_f + \alpha^*$. This implies a linear correlation between the scalar capability $P_m$ and the compute $\log(C)$ for models within a specific family on a downstream task (see empirical validation in Fig. E.3). Since $\theta_f$ and $\nu_f$ are unknown a priori, we can fit these coefficients $w_f, b_f$ via linear regression from $\log(C)$ to $P$ using models from the specific family $f$.

In the multi-model family case, we can map all models to a shared, FLOPs-based capability measure of a specific family $f$. The core idea is to represent each model's capabilities by the following hypothetical: "how many FLOPs ($\bar{C}_{m,f}$) would it take for a model in a family $f$ to match a model $m$". We call $\bar{C}_{m,f}$ the $f$-**equivalent FLOPs** for model $m$, as it represents the performance of model $m$ relative to models in the reference model family $f$. This measure can be computed fairly easily as

$$\log(\bar{C}_{m,f}) := \frac{1}{w_f^*} \left( \beta^{*\top} S_m + \alpha^* - b_f^* \right), \tag{8}$$

obtained from solving for $\log(C_m)$ in Eq. (7). Throughout the remainder of this work, we apply this scalar transformation where we pick Llama-2 [92] as the reference family $f$, and so the x-axis of all of our plots can be interpreted as "model capabilities, as measured in units of Llama-2 FLOPs".

## 4 Validating Observational Scaling Laws

We evaluate the usefulness of observational scaling laws by showing that they accurately predict the scaling behaviors of LMs over complex, hard-to-predict phenomena (like emergent phenomena and agentic abilities) and help estimate the value of techniques such as Chain-of-Thought.

To ensure that our scaling laws are actually predictive and that we are not simply overfitting through various choices in scaling law construction and hyperparameters, we design our experiments to have systematic holdout sets and robustness checks. We have also preregistered our predictions for *future* models after the initial release of the paper as a test of whether our scaling laws overfit current models. We release our code including the implementation and collected data at https://github.com/ryoungj/ObsScaling.

**Details in scaling law fits** For extracting PC measures, we fixed the number of PCs $K = 3$ as it covered $\sim 97\%$ of the variation in benchmark performance and it consistently yielded the best performance across most of our experiments, see Appx. E.4 for robustness checks on PC selection. For the capability-equivalent scale transformation, we used the Llama-2 [92] as the reference model family as it is currently the most representative and widely used open model in the community. For better interpretability and visualization, we used the accuracy metric, typically defined as $Y = 1 - E$, for fitting the scaling laws and making the plots.

**Holdout validation** To validate our observational scaling laws, our primary objective is to assess how accurately the scaling laws fit the available data and extrapolate from smaller-scale, less capable models to larger-scale, more powerful models. We validate this through systematic holdouts for the test set, where we split available models into weaker and stronger ones based on both scale or capability (e.g., FLOPs or accuracy). We used the weaker models to fit the scaling law and evaluated the extrapolated predictions on the stronger ones. To prevent any train-test leakage, all preprocessing steps (e.g., PCA imputation) were fitted on the train set only and then applied to the test set. Unless otherwise stated, we set the cutoff to include all models with training FLOPs less than or equal to that of Llama-2-7B ($8.4 \times 10^{22}$) as training data, resulting in a training set of 47 models and a test set of 30 models. We included robustness checks for different holdout strategies in Appx. E.4.

As baselines, we compare our scaling predictions to using existing compute-based scale measures like training FLOPs and model size. We used the mean squared error (MSE) on the test set as our main evaluation measure, which is comparable as the target range is always normalized (0 to 1).

**Preregistration of predictions** In the initial release of our paper (May 2024), we have preregistered our scaling predictions for future models (see preregistered functional forms in Appx. E.9) and committed to updating the manuscript on ArXiv with our prediction results after 4 months. We have assessed these predictions on new models released after the initial paper release, collected as of September 1st 2024, including most capable open models to date such as Llama 3.1-405B [24] and Qwen2-72B [105] (see the full collected model list in Appx. D.1.1), resulting in an additional test set of 20 models for robustness checks. The results are included in Fig. 4, and additional results on other tasks and new benchmarks are included in Appx. E.3.

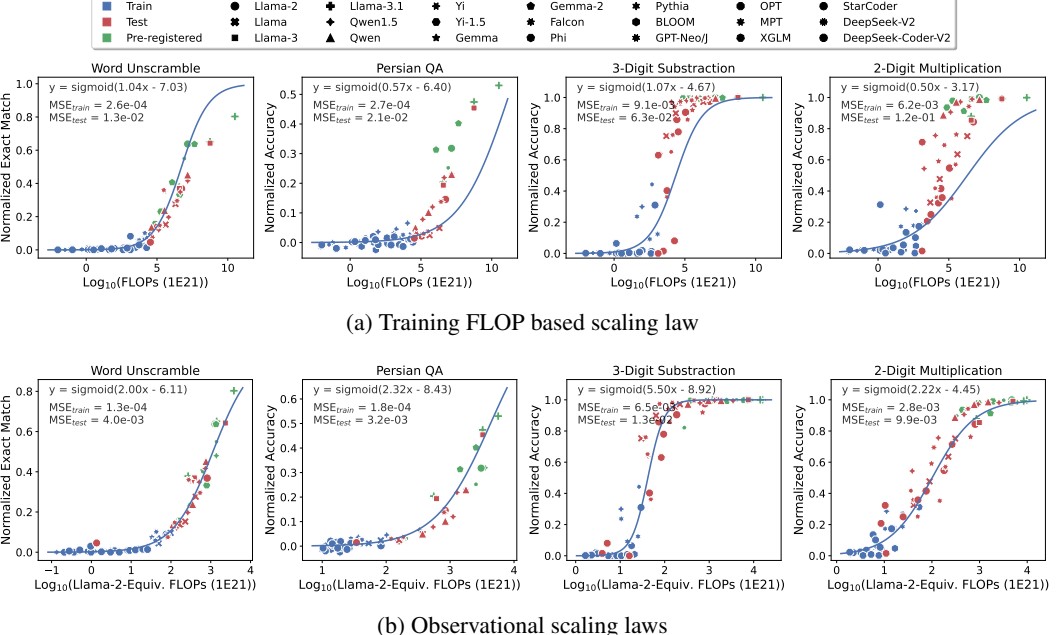

(a) Training FLOP based scaling law

(b) Observational scaling laws

Figure 4: "Emergent" capabilities of LMs can be accurately predicted from weaker models to stronger ones with observational scaling laws, and using PC measures as the predictor provides much more accurate predictions than using compute measures like training FLOPs and model size (see Fig. E.12). Our preregistered predictions also accurately extrapolate to new models released after the initial paper release, including Llama-3.1-405B [24]. Four tasks from BigBench [82], which are identified as "emergent" in [98], are used for illustration.

## 4.1 Predictability of "Emergent" Capabilities

Recent works have argued that many LM capabilities are "*emergent*" and cannot easily be predicted from small-scale models [27, 98]. There have been ongoing debates about whether these capabilities are truly discontinuous and whether the discontinuity is an artifact of the metric used [23, 39, 60, 78] or lack of high-resolution data points [38]. The debate has been complicated by the fact that existing scaling analyses (including the original ones in Wei et al. [98]) have very few points [38]. When there are only 5 models across many orders of magnitudes of scale, phenomena can appear to be discontinuous, even if the underlying phenomenon is a smooth but rapidly varying sigmoid.

We show that the higher resolution of observational scaling laws allows us to clearly see smooth sigmoidal curves in phenomena that were identified as emergent in Wei et al. [98], and even more surprisingly, we can often accurately forecast the transition points where models go from near-random to high performance using only models whose performance is only slightly above random. Our findings validate the observational approach to scaling laws and provide evidence that higher-resolution scaling laws could help us better understand scaling phenomena for LMs.

**Setup** We tested on four BigBench [82] tasks that were labeled as "emergent" in Wei et al. [98], including two arithmetic tasks (3-digit subtraction and 2-digit multiplication) and two non-arithmetic tasks (word unscramble and Persian QA). Additional results on more tasks covering Wei et al. [98] are included in Appx. E.5. For the models, we included base pretrained models following the approach of Wei et al. [98]. For non-arithmetic tasks, we used the default FLOPs cutoff. For arithmetic tasks, we found that this cutoff resulted in an excess of training data near perfect performance (see results in Fig. E.13), making the prediction tasks trivial. Consequently, we reduced the cutoff to a quarter of the default value and also excluded GSM8K (which may be a superset of arithmetic tasks) from our base metrics $B$ to make the tasks more challenging.

**Prediction results** Fig. 4 shows our prediction results using our PC measures as well as the baseline of predicting performance based on training FLOPs. We find that these capabilities can be accurately predicted using our PC measures, even when only using models that perform poorly. In contrast, using training FLOPs results in significantly poorer extrapolation on the test set and fits on the train set, as indicated by the much higher MSE values. This discrepancy is likely due to the incomparability of training FLOPs across different model families. Additional results of the model size baseline are included in Appx. E.5.

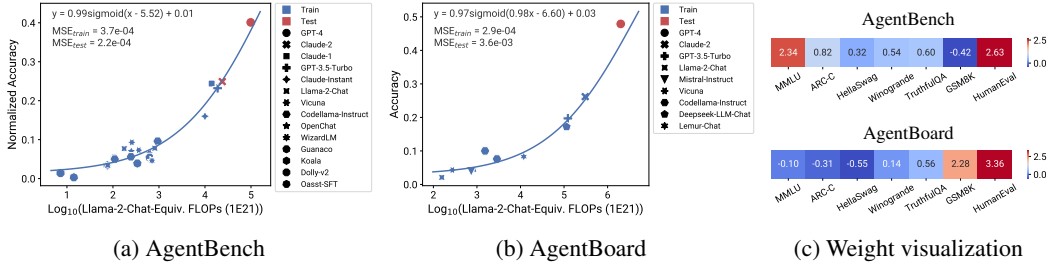

|               |               |               |
| ------------- | ------------- | ------------- |
| (a) AgentBench | (b) AgentBoard | (c) Weight visualization |

Figure 5: (a)-(b) The agentic capabilities of instruction-tuned LMs measured by agent benchmarks can be accurately predicted from weaker models (sub GPT-3.5) to stronger ones (e.g., GPT-4) by their PC measures. (c) The fitted weights ($\beta^\top\gamma$) on both benchmarks demonstrate the importance of programming capabilities (HumanEval) for the agentic capabilities of LMs.

## 4.2 Predictability of Agentic Capabilities

There is significant interest in building autonomous agents using LMs, with notable examples including AutoGPT [75], Devin [46], and SWE-agent [106]. Although the performance of these agents still falls far below human-level on challenging real-world tasks [43, 62, 110], there is a belief that future models at larger scales will significantly enhance these agents' capabilities. However, there is a significant uncertainty about whether existing models that are trained for language and code capabilities will transfer well to agentic tasks that require taking actions over many rounds. In this section, we utilize our observational scaling laws to analyze the scaling properties of LMs' agentic capabilities w.r.t. their backbone model capabilities and show that agent performance is highly predictable from simple benchmark metrics.

**Setup** We tested on two standardized agent evaluation benchmarks, AgentBench [57] and Agent-Board [61], each is a collection of diverse tasks for evaluating LMs' agentic capabilities. For both benchmarks, we utilized their provided aggregated metrics on all tasks for prediction. Specifically, we used the "Overall Score" on AgentBench, which is a weighted average of scores across all tasks (denoted as "OA" there), and the "Average Success Rate" on AgentBoard. We included models that have been evaluated on each benchmark, which encompasses both open instruction-tuned models like LLaMA-2-Chat [92], and proprietary models like GPT-4 [66] and Claude-2 [3], see Table D.2 for a complete list of models. We followed the same procedure to collect standardized benchmark metrics $B$ for instruction-tuned models, see Appx. D.1.2 for details. Notably, since compute scale measures are not available for proprietary models, only our observational scaling laws apply here and not compute scaling laws. The default FLOPs cutoff does not apply either, and thus we held out the top 10% performing models on each agent benchmark as the test set to simulate weak-to-strong predictions, which included GPT-4 and Claude-2 on AgentBench and GPT-4 on AgentBoard.

**Prediction results** Fig. 5 illustrates the prediction results with our observational scaling laws using PC measures. We find that on both agent benchmarks, the performance of held-out models (GPT-4/Claude-2) can be accurately predicted from models with much weaker performance (> 10% gap). This indicates that the more complex agentic capabilities of LMs are well-correlated with and predictable from their base model capabilities, suggesting the promising scaling properties of LM-based agent capabilities as backbone LMs continue to scale up.

**Interpreting the capability dimensions** In Fig. 5c, we visualize the weights assigned to the base evaluation metrics on both benchmarks, which are derived from the regression weights fitted on PC measures and applied with learned PCA transformation, i.e., $\beta^\top\gamma$. We observe that the fitted weights assign significant importance to programming capabilities (HumanEval) on both benchmarks, underscoring its significance in defining the agentic capabilities of LMs. The weights also emphasize general knowledge (MMLU) on AgentBench, and reasoning capabilities (GSM8K) on AgentBoard, suggesting that these capabilities may also be important for LMs' agentic capabilities.

## 4.3 Predicting the Impact of Post-Training Techniques

When researchers propose a new prompting or post-training technique to improve a pretrained model, how can we know whether these gains will persist across models and scales? Systematic scaling analyses have been rare due to the small number of models within a single model family. Moreover, some recent works have argued that certain interventions, such as Chain-of-Thought [99], behave in an emergent way that is not predictable from smaller models [98]. Using observational scaling laws, we show that it is possible to make relatively accurate predictions on the effectiveness of techniques

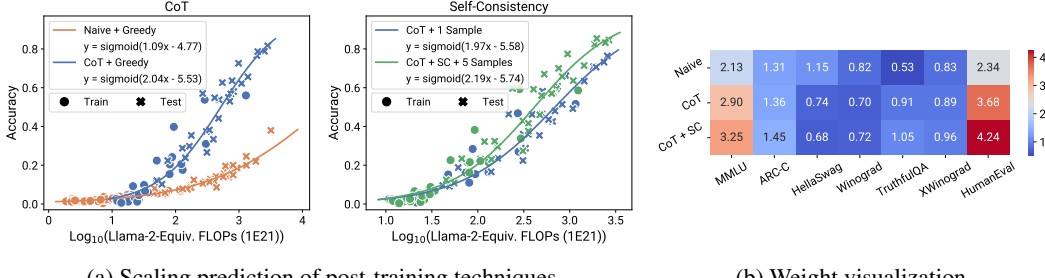

(a) Scaling prediction of post-training techniques      (b) Weight visualization

Figure 6: (a) The LM performance with and without techniques like CoT and Self-Consistency can be accurately predicted with observational scaling laws. The fitted scaling curves indicate that CoT has a better scaling behavior than SC. See Fig. E.15 for detailed per-method scaling plots and comparison with compute baselines. (b) The fitted weights ($\beta^\top \gamma$) demonstrate a very different pattern when CoT is applied, emphasizing general knowledge (MMLU) and programming capabilities (Humaneval).

such as Chain-of-Thought (CoT) [99] and Self-Consistency (SC) [97] as model scale increases. We focus on these post-training interventions in particular, as they are sometimes discussed as examples of post-training interventions that require scale to be effective [98, 99].

Our approach to quantifying the scaling properties of post-training is straightforward: we fit one observational scaling law using base model performance on a target benchmark (e.g., GSM8K few-shot), and then fit another on the performance of models with the post-training intervention (e.g., GSM8K w/ CoT). Each of these fits produces a sigmoidal scaling curve as a function of $\log(\bar{C}_f)$, and the relative gaps as a function of $\log(\bar{C}_f)$ indicates the scaling efficiency of the intervention.

**Setup** We tested on GSM8K with CoT and SC as post-training techniques and included additional results on BigBench-Hard [83] with CoT in Appx. E.6. As with our study on emergent phenomena on arithmetic tasks, we excluded GSM8K from the base metrics $B$ to avoid making the prediction tasks trivial. We included all the pretrained base models listed in Table D.1 including those specifically trained for code data and applied the default FLOPs cutoff for holdout validation. For CoT, we followed Wei et al. [99] and compared CoT prompting using eight reasoning examples with naive prompting using only few-shot examples in the greedy decoding setting. For SC, we sampled five CoT reasoning paths at temperature 0.7 to aggregate the final answers following Wang et al. [97] and compared it with a single sampled CoT answer.

**Prediction results** Fig. 6a shows the scaling predictions for CoT and SC using observational scaling laws. We find that the performance with (CoT, CoT + SC) and without (Naive) post-training techniques for stronger, larger scale models can be accurately predicted from weaker, smaller scale models. In contrast, predictions based on compute scale measures like model size and training FLOPs are less reliable as seen in Fig. E.15. Notably, the scaling trends between the two techniques differ; CoT shows a much more pronounced scaling trend compared to Self-Consistency w/ CoT.

**Interpreting the capability dimensions** Another advantage of observational scaling laws over scaling laws constructed on single families is that we can visualize the capabilities that are important to the post-training intervention. Fig. 6b visualizes the fitted regression weights $\beta$, mapped to the space of base capability benchmarks $B$ via $\beta^\top \gamma$. We clearly see that when we go from Naive to CoT, there are significantly higher weights placed on MMLU and HumanEval - meaning that scaling models in a way that enhances general knowledge (MMLU) and code (HumanEval) leads to greater gaps between CoT and the baseline, while improving along commonsense, such as Winogrande does not necessarily lead to improvements at scale. These analyses can inform how different post-training interventions affect different scaling recipes – such as code models vs general-purpose LLMs.

## 5   Selecting Low-Cost Model Subsets for Practical Scaling Analyses

Although our observational scaling law incurs no training cost, it still requires evaluating our benchmarks and post-training methods on a larger number of models. To make observational scaling analyses more broadly accessible, we identify a small set of representative models that maintain high prediction accuracy while significantly reducing the evaluation cost.

**Method** More specifically, we consider the constrained optimization problem of identifying the optimal set of models to choose for a regression problem, subject to the constraint that we select a model subset $\mathcal{M}$ of at most $M_{\max}$ models from the set of all models $\mathcal{M}_a$. To define optimality, we turn

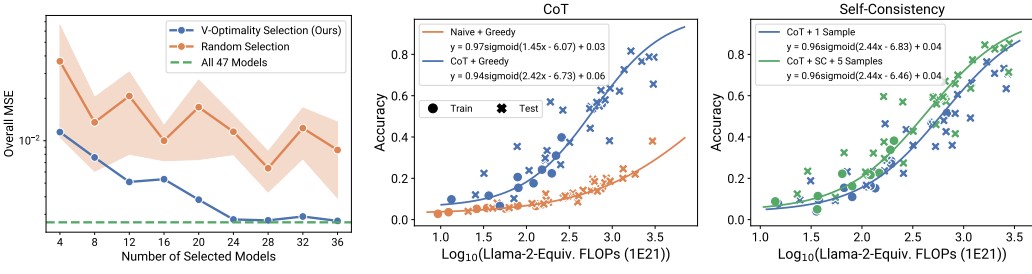

(a) Prediction error vs model counts

(b) Prediction results with only 12 models chosen by V-optimality

Figure 7: (a) Selecting the model subsets with our V-optimality criterion leads to significantly lower errors than random selection, and quickly converges to the errors of using the full set of models. (b) Using 12 (out of 47) models selected by our method maintains the overall prediction accuracy.

to the theory of optimal experimental design, which states that for linear regression with a fixed design $X$ and subset $\mathcal{M}$, the expected prediction error from using the subset $X_{\mathcal{M}}$ is $\mathrm{Tr}(X^{\top}X\left(X_{\mathcal{M}}^{\top}X_{\mathcal{M}}\right)^{-1})$. This gives a straightforward objective achieving the *V-optimality* [70]:

$$\min_{\mathcal{M}\in\mathcal{P}(\mathcal{M}_a)\ \mathrm{s.t.}|\mathcal{M}|\leq M_{\max}} \mathrm{Tr}(S^{\top}S\left(S_{\mathcal{M}}^{\top}S_{\mathcal{M}}\right)^{-1}) \tag{9}$$

where $S \in \mathbb{R}^{M \times K}$ is the model-capability matrix obtained from our PC analysis. We conduct a structured, exhaustive search over the 21 model families where we include or exclude entire model families under the budget constraint, as we believe these selected models are more interpretable.

**Validation**   We followed the setup in Sec. 4.3 for validating our selection method, as this represents the most likely application scenario for our observational scaling laws by practitioners. Our objective is to replicate our scaling analysis (using a full set of 47 models) in Fig. 6a using a small subset of models selected by our method. In Fig. 7a, we compute the geometric average of test MSEs on all prediction tasks (Naive, CoT, CoT + SC) as the evaluation metric for different selection methods. We find that our V-optimality selection method significantly outperforms random selection and quickly converges to the prediction performance of using the full set of models. In Fig. 7b, we show that using only a small subset of 12 models selected by our method, the fitted scaling curves already effectively capture the scaling trends of different post-training methods, in contrast to randomly selected models (Fig. E.18). To facilitate future scaling analyses at a low cost, we provide a reference list of models selected with our method under different budget constraints in Table E.1.

## 6   Conclusion, Limitations, and Future Work

We have presented observational scaling laws that generalize existing compute scaling laws to handle multiple model families using a shared, low-dimensional capability space. Using this approach, we show that we can build low-cost, high-resolution, and broad-coverage scaling laws that allow us to make accurate predictions for many complex scaling phenomena, such as emergent behaviors, agentic capabilities, and the value of post-training interventions. We provide concrete practical prescriptions for practitioners to perform similar scaling analyses in the hopes of encouraging more quantitative, scaling-law-based approaches to designing benchmarks and post-training methods.

**Limitations and future work**   Finally, we discuss some limitations of our approach and findings: Firstly, observational scaling laws are primarily applicable to post-training scaling analyses and do not directly translate to pretraining scenarios in the same way as standard compute-based scaling laws. Secondly, our study mostly focuses on the scaling behavior of model capabilities measured through few-shot prompting or basic prompting techniques (such as CoT, self-consistency, or simple agent scaffolding). Extending our approach to other post-training setups, including scenarios involving fine-tuning or more intensive inference-time computation [12, 81], would be valuable. Thirdly, while we have demonstrated that our observational scaling analyses can provide meaningful insights into improving particular models' complex capabilities, a promising direction for future work would be to apply the findings from our approach, such as by deriving surrogate measures for model complex capabilities that can be used to optimize models directly and efficiently. Lastly, our assumptions do not account for potential benchmark contamination (where particular benchmark data leaks into model training) or the heterogeneity within model families (where models within the same family may have varying compute efficiencies and scaling behaviors). Investigating the impact of these assumptions on our approach would be an interesting avenue for future research.

## Acknowledgements

We thank Zitong Yang for his assistance with an early experiment of the project. We also thank Jimmy Ba, Yann Dubois, Honghua Dong, Pavan Kapanipathi, Lisa Li, Karthik Narasimhan, Ethan Perez, Chenglei Si, Tristan Thrush, Zitong Yang, Shunyu Yao, the Hashimoto Group, and anonymous reviewers for their helpful discussions or feedback on the paper draft. This project is not possible without the open-source contributions including HuggingFace, EleutherAI LM Eval Harness [28], Open LLM Leaderboard [8], EvalPlus [55], vLLM [45], LMSys Chatbot Arena Leaderboard [18], and AlpacaEval Leaderboard [49].

TH and YR were supported in part by gifts from the Tianqiao and Chrissy Chen Institute, Open Philanthropy, Amazon ARA, Meta, and IBM. Resources used in preparing this research were provided in part by the Province of Ontario, the Government of Canada through CIFAR, and companies sponsoring the Vector Institute. We acknowledge the support of the Natural Sciences and Engineering Research Council of Canada (NSERC), RGPIN-2021-03445.

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

# A  Algorithm

In Algorithm A.1, we include the detailed algorithm for fitting the observational scaling laws as described in Sec. 3.

---

**Algorithm A.1:** Fitting observational scaling laws

---

**Args:** number of models $M$, number of LM benchmarks $T$, number of principal components $K$, reference model family $f$

**Input:** base LM benchmark error metrics $B \in \mathbb{R}^{T \times M}$, target downstream error metric $E \in \mathbb{R}^M$, LM compute scales $C \in \mathbb{R}^M$

**Result:** functional form of fitted scaling law $F$

/* Extract principal capability measures with applicable metric preprocessing                              */
$B \leftarrow \text{PCAImpute}(B)$                                    ▷ Fill in missing values with PCA imputation
$E \leftarrow \text{Normalize}(E)$                        ▷ Normalize metric to $[0, 1]$ for sigmoid non-linearity
$\gamma, S \leftarrow \text{PCA}(B, K)$                ▷ Fit PCA transformation $\gamma \in \mathbb{R}^{K \times T}$ and extract top $S = \gamma B$

/* Fit a non-linear regression with weights $\beta \in \mathbb{R}^K$ and bias $\alpha \in \mathbb{R}$, and sigmoidal scale $h \in \mathbb{R}$ */
$\beta^*, \alpha^*, h^* \leftarrow \text{Fit}\left(E = h\sigma(\beta^\top S + \alpha)\right)$                              ▷ Obtain optimal parameters
$P \leftarrow \beta^{*\top} S + \alpha^*$                        ▷ Obtain aggregated capability measures $P \in \mathbb{R}^M$

/* Project to the capability-equivalent scale of a reference model family                                   */
$w^*, b^* \leftarrow \text{Fit}(P_f = w\log(C_f) + b)$   ▷ Fit linear projection with models in the reference family
$\log(\bar{C}_f) \leftarrow (P - b^*)/w^*$                        ▷ Compute $f$-equivalent FLOPs for all models

/* Return the fitted scaling law with capability-equivalent scale transformation                           */
**return** $F : B \rightarrow h^*\sigma\left(\beta^{*\top}\gamma B + \alpha^*\right)$  *or*  $\bar{C}_f \rightarrow h^*\sigma\left(w^*\log(\bar{C}_f) + b^*\right)$

---

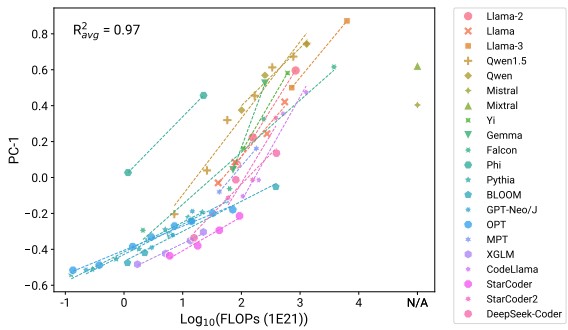 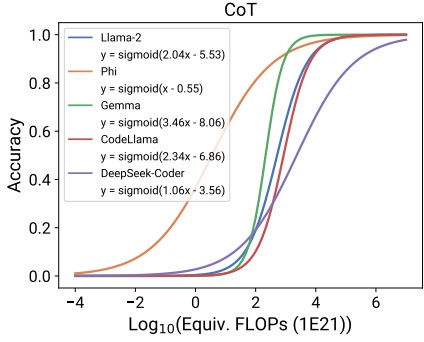

Figure B.1: PC-1 provides a smooth capability measure with a wider dynamic range than specific benchmarks like MMLU (Fig. E.4). In contrast to compute scale measures, it also enables the comparison of models from heterogeneous sources on a unified scale.

Figure B.2: By transforming the fitted scaling curves to $f$-equivalent scales for different model families, we can compare their scaling properties with CoT and analyze the effect of training recipes on the scaling behavior.

# B  Discussion and Other Applications of Observational Scaling

Our work validates the hypothesis that there is a low-dimensional space of LM capabilities that captures their scaling behaviors and can be measured via a low-rank decomposition of existing LM benchmarks – which interestingly connects to the item response theory in psychometrics [58] that models humans' test performance by their fundamental abilities such as general intelligence. While the majority of our work focuses on applications to scaling laws and predictions, we also find that the shared, low-dimensional capabilities could potentially be used as an evaluation metric and optimization target for LMs. We discuss some of these possibilities here.

**PC-1 as a smooth capability measure with high dynamic range**  Many existing benchmarks suffer from a limited dynamic range: they either saturate quickly for large models (e.g., HellaSwag, Winogrande) or have completely random performance for small models (e.g., MMLU, GSM8K), see Fig. E.4 for the behavior of each benchmark. In contrast, we find that PC-1 is a *smooth* capability measure that can be used to compare LMs across *many* (at least 5) orders of magnitude. This allows us to compare models from heterogeneous sources and of extremely different capabilities on a single, unified scale (Fig. B.1). We believe that the high dynamic range of PC1 may make it suitable as an optimization target for pretraining, where architecture or data interventions can be benchmarked against PC-1 at small scales and validated at large scales.

**Training data efficiency measurements using PC-1**  Extending these ideas further, since PC-1 serves as a unified measure of capabilities, it may serve as a good way to compare compute efficiencies across many model families. In Fig. B.1, we plot PC-1 against log-FLOPs and find that most models fall along a clear pattern in the training-compute to capabilities tradeoff curve. The Phi family is a clear outlier in compute efficiency, though this is likely because we are not accounting for the fact that Phi uses additional inference FLOPs to generate training data that is not shown in this figure.

**Post-training interventions and their interactions with model families**  Finally, we can analyze the interactions between post-training techniques and model families by projecting the fitted scaling curves in Fig. 6a to $f$-equivalent FLOPs for different families $f$ using Eq. (8). We can then identify which model families benefit the most from these techniques and the point at which they start to benefit. Fig. B.2 shows an example of comparing the predicted scaling of CoT across model families. We find that LMs benefit similarly from CoT, but that Phi is once again an outlier in its behavior: it benefits from CoT much earlier than other model families, but scales less rapidly. Similarly, models specifically trained on code (DeepSeek-Coder), also demonstrate an earlier transition but less rapid scaling compared to models trained with standard protocols. The distinct behavior of Phi/DeepSeek-Coder relative to other models indicates the importance of pretraining data in determining model scaling behaviors. While we did not specifically focus on these types of analysis in this work, we hope that our approach enables future works to gain further insights into differences between LM training recipes and their scaling behavior.

# C Extended Related Work

**Compute scaling laws** In standard scaling laws [6, 34–37, 44, 65], the "scale" is defined by the compute resources allocated to training LMs, such as the number of training FLOPs $C$, model parameters $N$, and training tokens $D$. Scaling laws are typically formulated as a power-law relationship between LMs' cross-entropy loss $L$ and their compute scale measures. Common functional forms include $L(N, D) = \frac{a}{N^\alpha} + \frac{b}{D^\beta} + e$ [37, 65] or $L(C) = \frac{c}{C^\gamma} + h$ [34, 44], where $C \approx 6ND$ [44] for the Transformer [93]. The parameters $\{\alpha, \beta, a, b, e\}$ or $\{\gamma, c, h\}$ are fitted by training LMs across different compute scales, varying $N$ and/or $D$, and measuring their loss. Our work differs from compute scaling laws in our goals – compute scaling aims to understand the scaling properties of pretraining, and thus focuses on a single model family and relates downstream performance to directly controllable quantities such as training compute. In contrast, we are interested in scaling laws for downstream, post-training performance, which leads us to consider scaling laws across model families and use more directly observable capability measures than compute.

**Downstream scaling laws** Scaling laws have been generalized beyond pretraining loss to analyze transfer learning [1, 35, 85] and downstream performance [15, 30, 34] across various domains, see Villalobos [94] for a comprehensive review. In particular, there has been evidence suggesting that the few-shot performance of LMs on downstream benchmarks is closely tied to compute measures like model size [13], but whether this is predictable with scaling laws remains debated. Extensive research has explored the difficulties of predicting benchmark performance due to their appearing rapid "emergence" [27, 83, 98], while recent works argued the discontinuity is due to the metrics used [60, 79] or the lack of data points [38] (see Anwar et al. [4] for a survey on this topic). Finnveden [25] and Owen [67] have investigated the use of linear and sigmoidal scaling laws, derived from pretraining loss or computational measures, to extrapolate the benchmark performance. Notably, Owen [67] also utilized publicly available LMs from different families to fit their compute scaling laws despite the potential discrepancies in their compute efficiencies. Recent studies have also more extensively investigated the correlations between the pretraining loss and downstream performance of LMs [39, 101], aiding in the understanding of downstream scaling [26] and emergent capabilities [23] of LMs. On the theory front, Arora and Goyal [5] derived a theory characterizing how performance on complex skills of LMs can be derived as a composition of base skills. While our work shares similar goals in that we aim to understand the downstream, post-training performance of models, we differ in our approach in that we aim to build practical higher-resolution scaling laws using multiple model families and their observable standard benchmark metrics.

**Correlations between benchmarks** Numerous works have investigated the correlations between different benchmarks across various contexts. Extensive research has explored the relationship between the out-of-distribution performance and in-distribution performance of machine learning models [63, 72, 73, 84, 104]. In the realm of NLP and LM benchmarks, Qiu et al. [71], Torregrossa et al. [90] found that different evaluations and metrics for word embeddings are highly correlated, and Liu et al. [56] observed a strong correlation between question-answering benchmarks. Moreover, Perlitz et al. [68], Polo et al. [69] observed strong correlations between samples within various LM benchmarks and utilized this observation to develop more efficient benchmarks. Most relevant to our work, Ilić [40] found that a single factor explains 85% of the performance on the Open LLM Leaderboard [8] and GLUE leaderboard [95], while Burnell et al. [14] extracted three factors for LM capabilities that account for 82% of the variation on the HELM benchmark [51], aligning with our observations. Our work also observes such benchmark correlations and low-rank structures but is unique in utilizing these properties for the purpose of scaling predictions that can be used directly for benchmark and algorithm development.

# D   Experimental Details

## D.1   Model Collection & Evaluation

### D.1.1   Pretrained Base Models

**Model collection**    We collected a broad set of representative open LMs covering 21 model families and a total of 77 models. These model families include Llama-2 [92], Llama [91], Llama-3 [24], Qwen1.5 [88], Qwen [7], Mistral [41], Mixtral [42], Yi [107], Gemma [86], Falcon [2], Phi [50], Pythia [10], BLOOM [100], GPT-Neo/J [11], OPT [109], MPT [89], XGLM [54], CodeLlama [76], StarCoder [48], StarCoder2 [59], DeepSeek-Coder [31]. For preregistration test, we have collected an additional set of 20 models covering 8 families released after May 2024 and as of September 1st 2024, including Llama-3.1 [24], Qwen2 [105], DeepSeek V2 [80], Gemma-2, [87], Jamba [52], Yi-1.5 [107], etc. For each model, we collected their available metadata including the number of model parameters $N$ and the amount of pretraining tokens $D$ by analyzing papers and other public information. We then estimated the training FLOPs $C$ using the simple estimate of $C \approx 6ND$ [44] for each model. Note that for models that were continually pretrained on additional data such as CodeLlama, we used the sum of the pretraining tokens and the additional continual pretraining tokens to estimate $D$. See Table D.1 for the collected metadata of these models.

**Benchmark collection & evaluation**    We collected a set of diverse benchmarks that assess various LMs' capabilities, including MMLU [32], ARC-C [19], HellaSwag [108], Winogrande [77], GSM8K [20], TruthfulQA [53], and XWinogrande [64], HumanEval [16]. For MMLU, ARC-C, HellaSwag, Winogrande, GSM8K, and TruthfulQA, we primarily sourced results from the Open LLM Leaderboard[1] [8], with updates current as of May 6th, 2024. When there were missing benchmark results, we followed the standardized evaluation protocols of the Open LLM Leaderboard and used the LM Eval Harness [28] library to evaluate the LMs. For XWinogrande, we used the LM Eval Harness library to evaluate the models with 5-shot examples. For HumanEval, we primarily used the EvalPlus [55] library and followed their standardized protocols for evaluation, and sourced the results from the EvalPlus leaderboard[2] when available. We used the 'Base Tests' results provided by EvalPlus for all the models. See Table D.1 for all collected benchmark results.

### D.1.2   Instruction-Tuned Models

**Model collection**    We collected the set of instruction-tuned models that have been evaluated on the AgentBench [57] and AgentBoard [61] benchmarks. These include models like GPT [66], Claude [3], Llama-2-Chat [92], Codellama-Instruct [76], Mistral-Instruct [41], Vicuna [17], Deepseek-LLM-Chat [9], Lemur-Chat [103], OpenChat [96], WizardLM [102], Guanaco [22], Koala [29], Dolly-v2 [21], OpenAssistant [47]. We followed the same procedure in Appx. D.1.1 to collect the metadata of open models, while for proprietary models these metadata were not publicly available. Note that we only counted the pretraining tokens (and the continual pretraining tokens when applicable) for $D$ and excluded the data for instruction-tuning or additional finetuning, as these are typically only a small fraction of the total data and are nuanced to estimate due to the complexities in data curation for instruction-tuning. See Table D.2 for the collected metadata of these models.

**Benchmark collection & evaluation**    For instruction-tuned models, we also included standard LM evaluations such as MMLU [32], ARC-C [19], HellaSwag [108], Winogrande [77], TruthfulQA [53], GSM8K [20], and HumanEval [16], and we followed the same protocols in Appx. D.1.1 for evaluating open models. For proprietary models like GPT and Claude, it is more nuanced to evaluate them with a unified protocol (e.g., due to the lack of access to likelihood scores), so we collected the official results from their respective papers and documentation for all standard benchmarks (except for HumanEval, which we were able to evaluate using the EvalPlus library). Additionally, we collected Elo scores from the Chatbot Arena[3] [18] which assess instruction-following capabilities of these instruction-tuned models (as of February 2nd, 2024) for reference, we did not utilize this metric for our downstream predictions. See Table D.2 for all collected benchmark results.

## D.2   Downstream Evaluation

For all downstream tasks of pretrained base models included in Sec. 4.1 and Sec. 4.3, we used the LM Eval Harness [28] library to evaluate all the models. For the "emergent" capability tasks in Sec. 4.1, we applied likelihood-based evaluation [13] with 2-shot examples. For the post-training intervention

---

[1]`https://huggingface.co/spaces/HuggingFaceH4/open_llm_leaderboard`
[2]`https://evalplus.github.io/leaderboard.html`
[3]`https://huggingface.co/spaces/lmsys/chatbot-arena-leaderboard`

Table D.1: Collected metadata and base evaluation metrics for base pretrained models used in Sec. 4.1, Sec. 4.3, and Sec. 5. Model names follow the HuggingFace naming. See data collection details in Appx. D.1.1. For the most up-to-date results, please refer to `https://github.com/ryoungj/ObsScaling/blob/main/eval_results/base_llm_benchmark_eval.csv`.

| Model Family | Model | Param (B) | Data (T) | FLOPs (1E21) | MMLU | ARC-C | HellaSwag | Winograd | TruthfulQA | XWinograd | HumanEval |
|---|---|---|---|---|---|---|---|---|---|---|---|
| Llama-2 | Llama-2-7b-hf | 7.0 | 2.0 | 84.00 | 0.4380 | 0.5307 | 0.7774 | 0.7403 | 0.3898 | 0.7549 | 0.1280 |
| | Llama-2-13b-hf | 13.0 | 2.0 | 156.00 | 0.5434 | 0.5811 | 0.8097 | 0.7664 | 0.3417 | 0.7868 | 0.1829 |
| | Llama-2-70b-hf | 70.0 | 2.0 | 840.00 | 0.6983 | 0.6732 | 0.8733 | 0.8374 | 0.4492 | 0.8245 | 0.2988 |
| Llama | llama-7b | 6.7 | 1.0 | 40.20 | 0.3569 | 0.5094 | 0.7781 | 0.7143 | 0.3433 | 0.6932 | 0.1280 |
| | llama-13b | 13.0 | 1.0 | 78.00 | 0.4761 | 0.5614 | 0.8092 | 0.7624 | 0.3948 | 0.7304 | 0.1585 |
| | llama-30b | 32.5 | 1.4 | 273.00 | 0.5845 | 0.6143 | 0.8473 | 0.8003 | 0.4227 | 0.7711 | 0.2073 |
| | llama-65b | 65.2 | 1.4 | 547.68 | 0.6393 | 0.6348 | 0.8609 | 0.8256 | 0.4343 | 0.7768 | 0.2317 |
| Llama-3 | Meta-Llama-3-8B | 8.0 | 15.0 | 720.00 | 0.6649 | - | 0.8202 | 0.7711 | 0.4395 | 0.8012 | 0.3841 |
| | Meta-Llama-3-70B | 70.0 | 15.0 | 6300.00 | 0.7923 | - | 0.8798 | 0.8532 | 0.4556 | 0.8447 | 0.5244 |
| Qwen1.5 | Qwen1.5-0.5B | 0.5 | 2.4 | 7.20 | 0.3935 | 0.3148 | 0.4905 | 0.5722 | 0.3830 | 0.5756 | 0.1159 |
| | Qwen1.5-1.8B | 1.8 | 2.4 | 25.92 | 0.4671 | 0.3788 | 0.6142 | 0.6030 | 0.3943 | 0.6438 | 0.1829 |
| | Qwen1.5-4B | 4.0 | 2.4 | 57.60 | 0.5652 | 0.4846 | 0.7158 | 0.6622 | 0.4727 | 0.6888 | 0.2622 |
| | Qwen1.5-7B | 7.0 | 4.0 | 168.00 | 0.6197 | 0.5418 | 0.7851 | 0.7127 | 0.5108 | 0.7524 | 0.3476 |
| | Qwen1.5-14B | 14.0 | 4.0 | 336.00 | 0.6936 | 0.5657 | 0.8108 | 0.7348 | 0.5206 | 0.7775 | 0.3963 |
| | Qwen1.5-32B | 32.0 | 4.0 | 768.00 | 0.7430 | 0.6357 | 0.8500 | 0.8145 | 0.5739 | 0.7912 | 0.4207 |
| | Qwen1.5-72B | 72.0 | 3.0 | 1296.00 | 0.7720 | 0.6587 | 0.8599 | 0.8303 | 0.5961 | 0.8258 | 0.4512 |
| Qwen | Qwen-7B | 7.0 | 2.4 | 100.80 | 0.5984 | 0.5137 | 0.7847 | 0.7269 | 0.4779 | 0.7346 | 0.3171 |
| | Qwen-14B | 14.0 | 3.0 | 252.00 | 0.6770 | 0.5828 | 0.8399 | 0.7680 | 0.4943 | 0.7915 | 0.3537 |
| | Qwen-72B | 72.0 | 3.0 | 1296.00 | 0.7737 | 0.6519 | 0.8594 | 0.8248 | 0.6019 | 0.8287 | 0.3720 |
| Mistral | Mistral-7B-v0.1 | 7.3 | - | - | 0.6416 | 0.5998 | 0.8331 | 0.7861 | 0.4215 | 0.7819 | 0.2744 |
| Mixtral | Mixtral-8x7B-v0.1 | 45.0 | - | - | 0.7188 | 0.6638 | 0.8646 | 0.8169 | 0.4681 | 0.8002 | 0.3354 |
| Yi | Yi-6B | 6.0 | 3.0 | 108.00 | 0.6411 | 0.5555 | 0.7657 | 0.7419 | 0.4196 | 0.7239 | 0.1585 |
| | Yi-34B | 34.0 | 3.0 | 612.00 | 0.7635 | 0.6459 | 0.8569 | 0.8303 | 0.5623 | 0.7956 | 0.2683 |
| Gemma | gemma-2b | 2.0 | 6.0 | 72.00 | 0.4177 | 0.4838 | 0.7177 | 0.6630 | 0.3308 | 0.7093 | 0.2317 |
| | gemma-7b | 7.0 | 6.0 | 252.00 | 0.6603 | 0.6109 | 0.8247 | 0.7845 | 0.4491 | 0.7839 | 0.3354 |
| Falcon | falcon-rw-1b | 1.0 | 0.35 | 2.10 | 0.2528 | 0.3507 | 0.6356 | 0.6204 | 0.3596 | 0.5355 | - |
| | falcon-7b | 7.0 | 1.5 | 63.00 | 0.2779 | 0.4787 | 0.7813 | 0.7238 | 0.3426 | 0.7176 | - |
| | falcon-40b | 40.0 | 1.0 | 240.00 | 0.5698 | 0.6195 | 0.8528 | 0.8129 | 0.4172 | 0.7846 | - |
| | falcon-180B | 180.0 | 3.5 | 3780.00 | 0.6959 | 0.6920 | 0.8889 | 0.8690 | 0.4516 | 0.8446 | - |
| Phi | phi-1_5 | 1.3 | 0.15 | 1.17 | 0.4389 | 0.5290 | 0.6379 | 0.7222 | 0.4089 | 0.5111 | 0.3415 |
| | phi-2 | 2.7 | 1.4 | 22.68 | 0.5792 | 0.6101 | 0.7492 | 0.7348 | 0.4424 | 0.5267 | 0.4939 |
| Pythia | pythia-70m-deduped | 0.07 | 0.3 | 0.13 | 0.2526 | 0.2108 | 0.2717 | 0.4964 | 0.4751 | 0.5101 | 0.0000 |
| | pythia-160m-deduped | 0.16 | 0.3 | 0.29 | 0.2486 | 0.2406 | 0.3139 | 0.5138 | 0.4434 | 0.5236 | 0.0000 |
| | pythia-410m-deduped | 0.41 | 0.3 | 0.74 | 0.2599 | 0.2483 | 0.4129 | 0.5438 | 0.4095 | 0.5363 | 0.0122 |
| | pythia-1b-deduped | 1.0 | 0.3 | 1.80 | 0.2427 | 0.2910 | 0.4965 | 0.5359 | 0.3894 | 0.5610 | 0.0427 |
| | pythia-1.4b-deduped | 1.4 | 0.3 | 2.52 | 0.2556 | 0.3268 | 0.5496 | 0.5730 | 0.3866 | 0.5941 | 0.0427 |
| | pythia-2.8b-deduped | 2.8 | 0.3 | 5.04 | 0.2678 | 0.3626 | 0.6066 | 0.6022 | 0.3556 | 0.6400 | 0.0488 |
| | pythia-6.9b-deduped | 6.9 | 0.3 | 12.42 | 0.2648 | 0.4130 | 0.6705 | 0.6409 | 0.3519 | 0.6525 | 0.0854 |
| | pythia-12b-deduped | 12.0 | 0.3 | 21.60 | 0.2563 | 0.4138 | 0.7026 | 0.6646 | 0.3300 | 0.6824 | 0.1159 |
| BLOOM | bloom-560m | 0.56 | 0.341 | 1.15 | 0.2422 | 0.2474 | 0.3715 | 0.5193 | 0.4244 | 0.5786 | 0.0061 |
| | bloom-1b1 | 1.1 | 0.341 | 2.25 | 0.2670 | 0.2833 | 0.4278 | 0.5501 | 0.4180 | 0.6095 | 0.0000 |
| | bloom-3b | 3.0 | 0.341 | 6.14 | 0.2659 | 0.3575 | 0.5437 | 0.5762 | 0.4057 | 0.6648 | 0.0183 |
| | bloom-7b1 | 7.1 | 0.341 | 14.53 | 0.2625 | 0.4113 | 0.6200 | 0.6543 | 0.3890 | 0.6977 | 0.0488 |
| | bloom | 176.0 | 0.366 | 386.50 | 0.3085 | 0.5043 | 0.7641 | 0.7206 | 0.3976 | 0.7355 | 0.1220 |
| GPT-Neo/J | gpt-neo-125m | 0.125 | 0.3 | 0.22 | 0.2597 | 0.2295 | 0.3026 | 0.5178 | 0.4558 | 0.5022 | 0.0061 |
| | gpt-neo-1.3B | 1.3 | 0.38 | 2.96 | 0.2482 | 0.3123 | 0.4847 | 0.5691 | 0.3963 | 0.5611 | 0.0366 |
| | gpt-neo-2.7B | 2.7 | 0.42 | 6.80 | 0.2645 | 0.3336 | 0.5624 | 0.6006 | 0.3978 | 0.5740 | 0.0671 |
| | gpt-j-6b | 6.05 | 0.402 | 14.59 | 0.2678 | 0.4138 | 0.6754 | 0.6598 | 0.3596 | 0.6811 | 0.1159 |
| | gpt-neox-20b | 20.0 | 0.472 | 56.64 | 0.2500 | 0.4573 | 0.7345 | 0.6890 | 0.3161 | 0.7163 | 0.1280 |
| OPT | opt-125m | 0.125 | 0.18 | 0.14 | 0.2602 | 0.2287 | 0.3147 | 0.5162 | 0.4287 | 0.4987 | 0.0000 |
| | opt-350m | 0.35 | 0.18 | 0.38 | 0.2602 | 0.2355 | 0.3673 | 0.5264 | 0.4083 | 0.5181 | 0.0000 |
| | opt-1.3b | 1.3 | 0.18 | 1.40 | 0.2496 | 0.2952 | 0.5453 | 0.5975 | 0.3871 | 0.5440 | 0.0000 |
| | opt-2.7b | 2.7 | 0.18 | 2.92 | 0.2543 | 0.3396 | 0.6143 | 0.6196 | 0.3743 | 0.5685 | 0.0000 |
| | opt-6.7b | 6.7 | 0.18 | 7.24 | 0.2457 | 0.3916 | 0.6866 | 0.6598 | 0.3512 | 0.5943 | 0.0061 |
| | opt-13b | 13.0 | 0.18 | 14.04 | 0.2490 | 0.3993 | 0.7120 | 0.6851 | 0.3410 | 0.6088 | 0.0061 |
| | opt-30b | 30.0 | 0.18 | 32.40 | 0.2666 | 0.4326 | 0.7407 | 0.7064 | 0.3516 | 0.6264 | 0.0122 |
| | opt-66b | 66.0 | 0.18 | 71.28 | 0.2699 | 0.4633 | 0.7625 | 0.7001 | 0.3543 | 0.6426 | 0.0122 |
| MPT | mpt-7b | 7.0 | 1.0 | 42.00 | 0.2807 | 0.4770 | 0.7753 | 0.7214 | 0.3355 | 0.7144 | 0.1646 |
| | mpt-30b | 30.0 | 1.0 | 180.00 | 0.4800 | 0.5597 | 0.8242 | 0.7490 | 0.3842 | 0.7453 | 0.2134 |
| XGLM | xglm-564M | 0.564 | 0.5 | 1.69 | 0.2518 | 0.2457 | 0.3464 | 0.5225 | 0.4043 | 0.5855 | 0.0000 |
| | xglm-1.7B | 1.7 | 0.5 | 5.10 | 0.2510 | 0.2585 | 0.4568 | 0.5391 | 0.3721 | 0.6307 | 0.0000 |
| | xglm-4.5B | 4.5 | 0.5 | 13.50 | 0.2543 | 0.3148 | 0.5795 | 0.5493 | 0.3584 | 0.6585 | 0.0000 |
| | xglm-7.5B | 7.5 | 0.5 | 22.50 | 0.2779 | 0.3413 | 0.6077 | 0.5872 | 0.3666 | 0.6956 | 0.0000 |
| CodeLlama | CodeLlama-7b-hf | 7.0 | 2.52 | 105.84 | 0.3112 | 0.3993 | 0.6080 | 0.6401 | 0.3782 | 0.7297 | 0.3354 |
| | CodeLlama-13b-hf | 13.0 | 2.52 | 196.56 | 0.3281 | 0.4087 | 0.6335 | 0.6717 | 0.4379 | 0.7349 | 0.3841 |
| | CodeLlama-34b-hf | 34.0 | 2.52 | 514.08 | 0.5502 | 0.4854 | 0.7582 | 0.7356 | 0.3911 | 0.7861 | 0.4756 |
| | CodeLlama-70b-hf | 70.0 | 3.02 | 1268.40 | 0.5967 | 0.5674 | 0.7821 | 0.7522 | 0.3979 | 0.7756 | 0.5488 |
| StarCoder | starcoderbase-1b | 1.0 | 1.0 | 6.00 | 0.2667 | 0.2270 | 0.3431 | 0.4996 | 0.4579 | 0.5617 | 0.1460 |
| | starcoderbase-3b | 3.0 | 1.0 | 18.00 | 0.2735 | 0.2585 | 0.3911 | 0.5114 | 0.4305 | 0.5976 | 0.1770 |
| | starcoderbase-7b | 7.0 | 1.0 | 42.00 | 0.2845 | 0.2986 | 0.4387 | 0.5438 | 0.4046 | 0.5978 | 0.2440 |
| | starcoderbase | 15.5 | 1.0 | 93.00 | 0.3212 | 0.3029 | 0.4721 | 0.5580 | 0.4002 | 0.5952 | 0.3410 |
| StarCoder2 | starcoder2-3b | 3.0 | 3.3 | 59.40 | 0.3865 | 0.3456 | 0.4762 | 0.5454 | 0.4049 | 0.6037 | 0.3170 |
| | starcoder2-7b | 7.0 | 3.7 | 155.40 | 0.4121 | 0.3831 | 0.5191 | 0.5919 | 0.4199 | 0.6201 | 0.3540 |
| | starcoder2-15b | 15.0 | 4.3 | 387.00 | 0.5135 | 0.4735 | 0.6409 | 0.6385 | 0.3787 | 0.7383 | 0.4630 |
| DeepSeek-Coder | deepseek-coder-1.3b-base | 1.3 | 2.0 | 15.60 | 0.2602 | 0.2577 | 0.3928 | 0.5272 | 0.4261 | 0.6063 | 0.2870 |
| | deepseek-coder-6.7b-base | 6.7 | 2.0 | 80.40 | 0.3839 | 0.3703 | 0.5346 | 0.5809 | 0.4028 | 0.6789 | 0.4760 |
| | deepseek-coder-33b-base | 33.0 | 2.0 | 396.00 | 0.4091 | 0.4249 | 0.5999 | 0.6243 | 0.3997 | 0.6961 | 0.5120 |

tasks in Sec. 4.3, we used the same evaluation protocol as the original papers, as described in the main paper. For agentic capability tasks of instruction-tuned models in Sec. 4.2, we directly sourced the results from the AgentBench [57] and AgentBoard [61] leaderboards and scaled the metrics to [0, 1].

Table D.2: Collected metadata and base evaluation metrics for instruction-tuned models used in Sec. 4.2. Model names follow the HuggingFace naming for open models. See data collection details in Appx. D.1.2.

| Model Family | Model | Param (B) | Data (T) | FLOPs (1E21) | Arena-Elo | MMLU | ARC-C | HellaSwag | Winogrande | TruthfulQA | HumanEval |
|---|---|---|---|---|---|---|---|---|---|---|---|
| GPT | gpt-4-0613 | - | - | - | 1161.6608 | 0.8640 | 0.9630 | 0.9530 | 0.8750 | 0.5900 | 0.8720 |
| | gpt-4-0314 | - | - | - | 1189.5486 | 0.8640 | 0.9630 | 0.9530 | 0.8750 | 0.5900 | 0.9024 |
| | gpt-3.5-turbo-0613 | - | - | - | 1118.1123 | 0.7000 | 0.8520 | 0.8550 | 0.8160 | 0.4700 | 0.7744 |
| Claude | claude-2.0 | - | - | - | 1132.3173 | 0.7850 | 0.9100 | - | - | 0.6900 | 0.6707 |
| | claude-1.3 | - | - | - | 1149.3443 | 0.7700 | 0.9000 | - | - | 0.6200 | 0.6159 |
| | claude-instant-1.1 | - | - | - | 1109.4714 | 0.7340 | 0.8570 | - | - | 0.6600 | 0.5915 |
| Llama-2-Chat | llama-2-7b-chat | 7.0 | 2.0 | 84.00 | 1024.1411 | 0.4706 | 0.5290 | 0.7855 | 0.7174 | 0.4557 | 0.1220 |
| | llama-2-13b-chat | 13.0 | 2.0 | 156.00 | 1041.8442 | 0.5412 | 0.5904 | 0.8194 | 0.7451 | 0.4412 | 0.1829 |
| | llama-2-70b-chat | 70.0 | 2.0 | 840.00 | 1082.0000 | 0.6345 | 0.6459 | 0.8588 | 0.8051 | 0.5280 | 0.3171 |
| Codellama-Instruct | codellama-7b-instruct | 7.0 | 2.52 | 105.84 | - | 0.3454 | 0.3652 | 0.5544 | 0.6456 | 0.4125 | 0.3963 |
| | codellama-13b-instruct | 13.0 | 2.52 | 196.56 | - | 0.3889 | 0.4454 | 0.6493 | 0.6803 | 0.4588 | 0.4451 |
| | codellama-34b-instruct | 34.0 | 2.52 | 514.08 | 1043.4381 | 0.5462 | 0.5427 | 0.7692 | 0.7451 | 0.4444 | 0.4878 |
| Mistral-Instruct | mistral-7b-instruct-v0.1 | 7.0 | - | - | 1006.4716 | 0.5539 | 0.5452 | 0.7563 | 0.7372 | 0.5628 | 0.3537 |
| Vicuna | vicuna-7b-v1.5 | 7.0 | 2.0 | 84.00 | 1004.9595 | 0.5031 | 0.5324 | 0.7739 | 0.7214 | 0.5033 | 0.1341 |
| | vicuna-13b-v1.5 | 13.0 | 2.0 | 156.00 | 1040.3549 | 0.5624 | 0.5657 | 0.8109 | 0.7466 | 0.5107 | 0.2134 |
| | vicuna-13b-16k | 13.0 | 2.0 | 156.00 | - | 0.5489 | 0.5674 | 0.8037 | 0.7285 | 0.5196 | 0.2500 |
| | vicuna-33b-v1.3 | 33.0 | 2.0 | 396.00 | 1093.4174 | 0.5921 | 0.6160 | 0.8306 | 0.7703 | 0.5609 | 0.2134 |
| Deepseek-LLM-Chat | deepseek-llm-67b-chat | 67.0 | 2.0 | 804.00 | 1081.7334 | 0.7174 | 0.6775 | 0.8680 | 0.8421 | 0.5583 | 0.7012 |
| Lemur-Chat | lemur-70b-chat-v1 | 70.0 | 2.09 | 877.80 | - | 0.6599 | 0.6698 | 0.8573 | 0.8169 | 0.5658 | 0.5915 |
| OpenChat | openchat-13b-v3.2 | 13.0 | 2.0 | 156.00 | - | 0.5668 | 0.5964 | 0.8268 | 0.7695 | 0.4449 | 0.2073 |
| WizardLM | wizardlm-13b-v1.2 | 13.0 | 2.0 | 156.00 | 1058.0881 | 0.5367 | 0.5904 | 0.8221 | 0.7190 | 0.4727 | 0.3902 |
| | wizardlm-30b-v1.0 | 30.0 | 3.0 | 540.00 | - | 0.5888 | 0.6254 | 0.8327 | 0.7751 | 0.5249 | - |
| Guanaco | guanaco-33b | 33.0 | 1.4 | 277.20 | 1031.9123 | 0.5569 | 0.6246 | 0.8448 | - | 0.5122 | 0.2622 |
| | guanaco-65b | 65.0 | 1.4 | 546.00 | - | 0.6251 | 0.6544 | 0.8647 | 0.8240 | 0.5281 | 0.2744 |
| Koala | koala-13b | 13.0 | 1.0 | 78.00 | 965.7386 | 0.4501 | 0.5299 | 0.7759 | 0.7403 | 0.5023 | 0.1220 |
| Dolly-v2 | dolly-v2-12b | 12.0 | 0.3 | 21.60 | 822.6771 | 0.2581 | 0.4241 | 0.7253 | 0.6085 | 0.3383 | 0.0000 |
| OpenAssistant | oasst-sft-4-pythia-12b-epoch-3.5 | 12.0 | 0.3 | 21.60 | - | 0.2682 | 0.4573 | 0.6859 | 0.6590 | 0.3781 | 0.0793 |

## D.3   PCA Analysis

**PCA imputation**   The PCA imputation starts with a simple mean imputation for missing values in the data matrix, and then PCA is applied to transform the data into a lower-dimensional space where the missing values are imputed by the PCA reconstruction. The above procedure is repeated until the imputed values converge or reach a maximum of 1000 iterations. By default, we used the first principal component (PC-1) to impute the missing values, as we found it to be the most robust in our preliminary experiments. Notably, when there are train and test splits, we first applied the PCA imputation procedure on the training set and then applied the same transformation to the test set to prevent any train-test leakage.

**PC extraction**   When applying PCA to extracting the capability measures, we extracted the top $K = 3$ principal components from the model-capability matrix. By default, we mean-centered the data before applying PCA without additional scaling, since most evaluation metrics are already normalized into $[0, 1]$. Similar to PCA imputation, we only fitted the PCA on the training set and applied the same transformation to the test set to prevent any train-test leakage.

# E    Additional Results

## E.1    PC Analysis of Instruction-Tuned LMs

In Fig. E.1, we conducted a PC analysis for instruction-tuned models (see the model list in Table D.2) following exactly the same procedure as Fig. 2. We find that the extracted PC measures for instruction-tuned LMs follow similar patterns as pretrained models and exhibit an even more significant low-rank structure, with the top 3 PCs explaining about 98.6% of the variance in the benchmark performance.

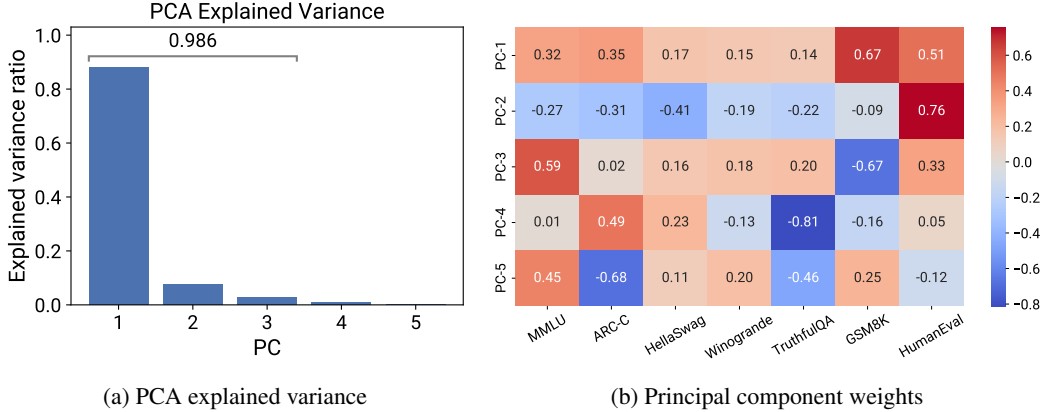

(a) PCA explained variance                    (b) Principal component weights

Figure E.1: The extracted PC measures for instruction-tuned LMs follow similar low-rank structures and interpretable patterns as pretrained base LMs (see Fig. 2).

## E.2    Properties of PC measures

**Lower-ranked PCs linearly correlate with log-compute measures**    In Fig. 3, we showed that the top PC-1 linearly correlates with log-compute scale measures (log-training FLOPs) within each comparable model family. In Fig. E.2, we show that this linear correlation generally holds for lower-ranked PCs, specifically PC-2 and PC-3, though the correlation tends to decrease with lower-rank PCs compared to the top PC-1.

**Aggregated PCs linearly correlate with log-compute measures**    When fitting our observational scaling laws, we utilized the (hypothetical) linear relation between the aggregated PC measures $P_m := \beta^{*\top} S_m$ and the log-compute measures $\log(C_m)$ within each model family to transform $P_m$ into compute-equivalent scales (Eq. (8)) . This linear correlation has been partially validated through the linear correlation of top PCs (Fig. 3 & Fig. E.2). Here we more directly validate this linearity by analyzing the aggregated PC measures $P_m$ fitted on specific tasks. Specifically, in Fig. E.3, we visualize the fitted $P_m$ on the "emergent" capability tasks (i.e., Fig. 4b) versus the compute measures $\log(C_m)$ within each comparable model family. We find that the aggregated PC measures generally exhibit a linear correlation with the log-compute measures within each family. Notably, the linear correlation is consistently significant for the Llama-2 family, which we have used as the default reference family for computing the equivalent scales in our experiments.

**Single benchmark metric suffers from limited dynamic range**    In Fig. B.1, we have shown that PC-1 can serve as a smooth capability measure for LMs that provide meaningful readouts across many orders of scales (about 5 orders of magnitude). In Fig. E.4, we show that using a single benchmark metric as LM capability measures amy suffer from a limited dynamic range. In particular, they may either saturate quickly for large models (e.g., HellaSwag, Winogrande) or provide random readouts for weak models (e.g., MMLU, GSM8K).

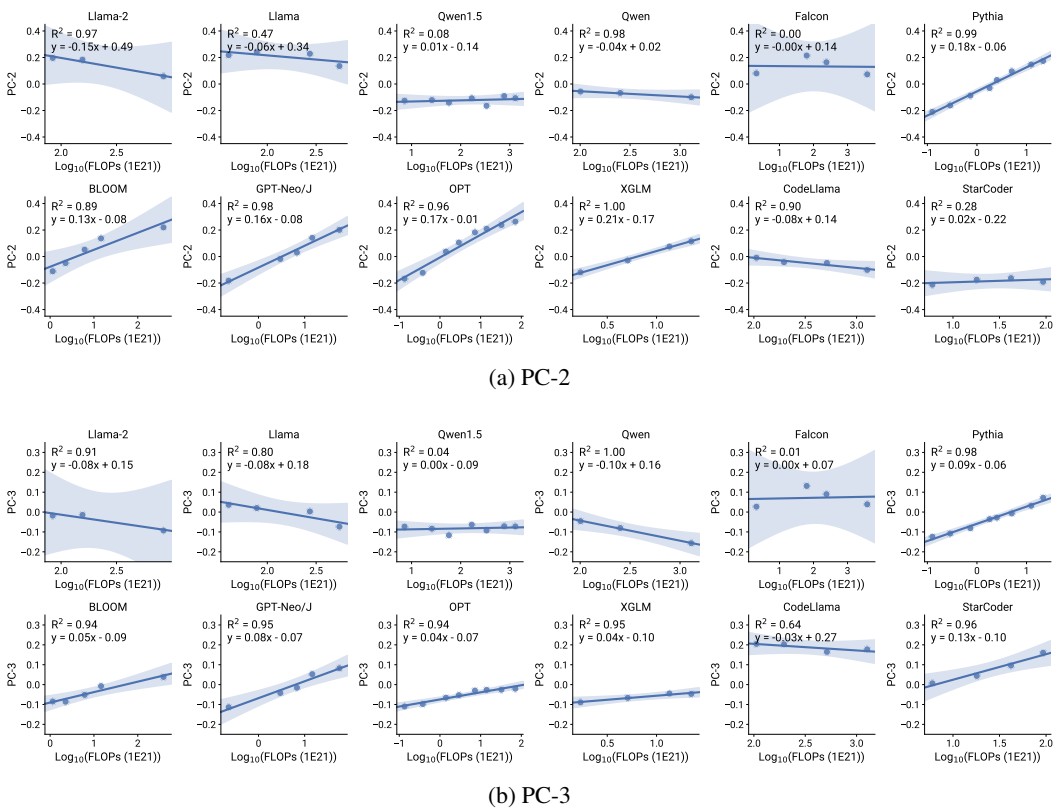

(a) PC-2

(b) PC-3

Figure E.2: The lower-ranked PC measures also linearly correlate with log-compute measures within each comparable model family, though the correlation decreases with lower-rank PCs.

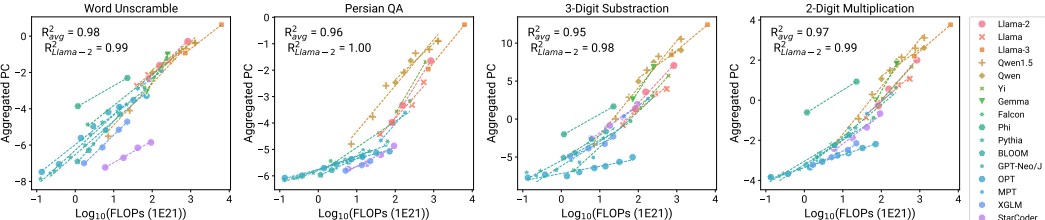

Figure E.3: The aggregated PC measures exhibit a strong linear correlation with the log-compute measures within each comparable model family, especially for Llama-2 which we have used as the default reference family for computing the $f$-equivalent FLOPs in our experiments.

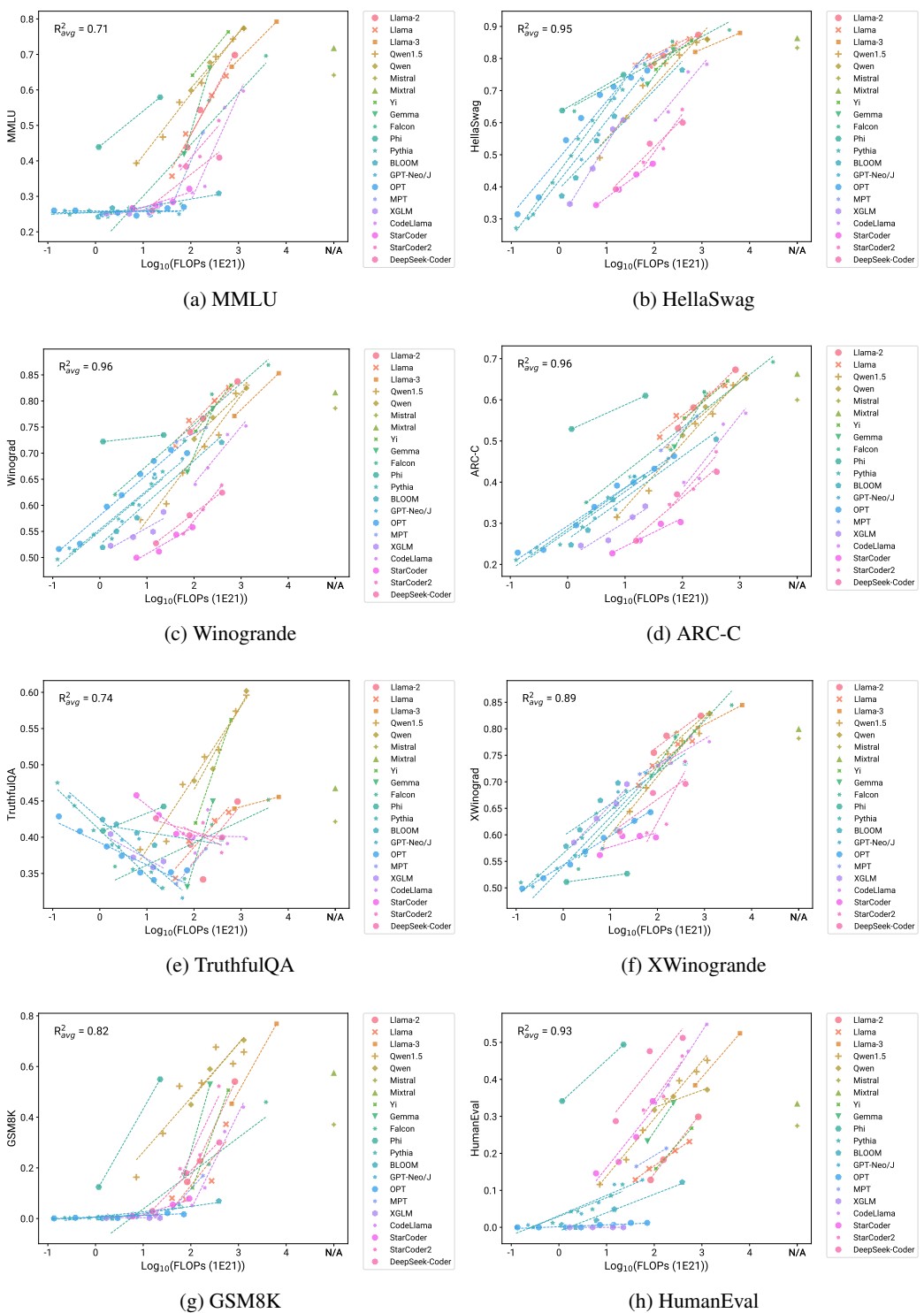

Figure E.4: Using a single benchmark metric to measure LM capabilities may suffer from a limited dynamic range. They may either saturate quickly for large models (e.g., HellaSwag, Winogrande) or provide random readouts for weak models (e.g., MMLU, GSM8K).

### E.3 Additional Preregisteration Results

**Preregistered predictions on post-training analysis tasks**   In Fig. E.5, we tested our preregistered predictions on the post-training analysis tasks. We observe reasonable forecasts on new models, and the predictions using PC measures outperform the ones using compute measures like training FLOPs.

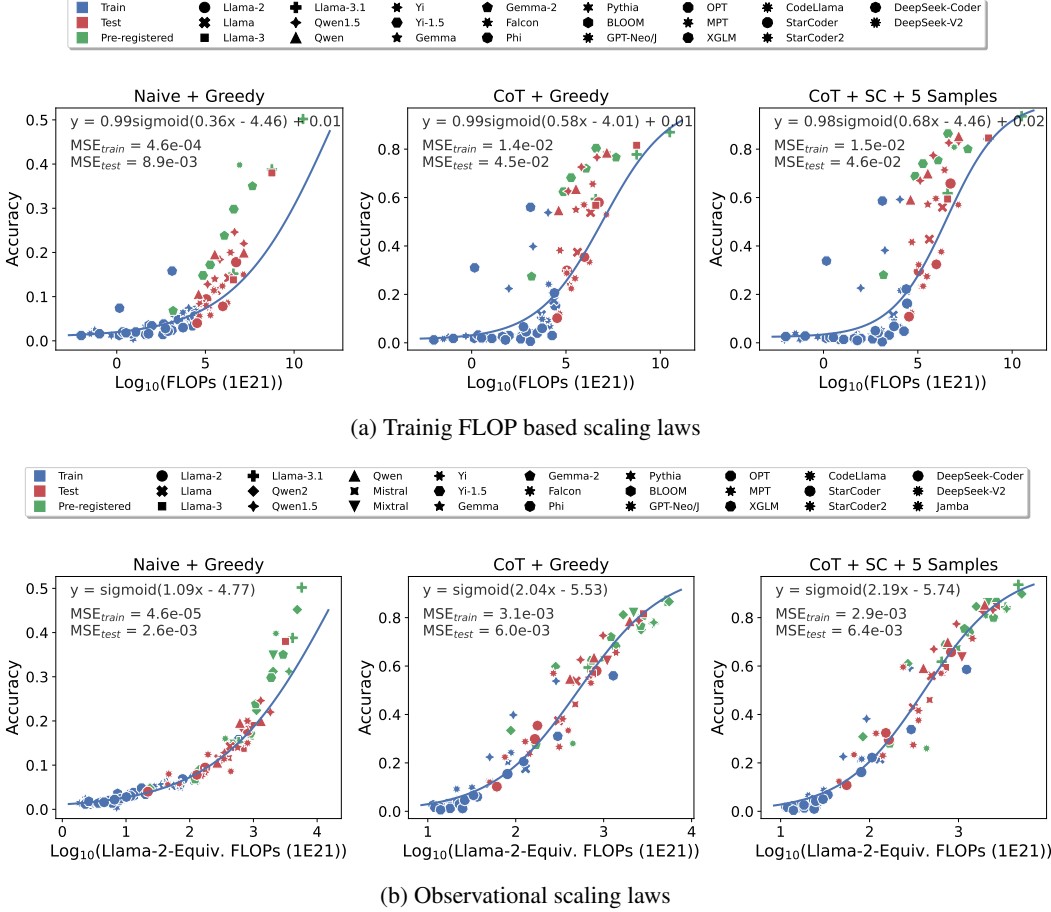

(a) Trainig FLOP based scaling laws

(b) Observational scaling laws

Figure E.5: Our preregistered predictions of observational scaling laws using PC measures (with # = 3) provides reasonable forecasts for new models that are released after our initial paper release. The predictions on naive prompting is a bit off, but still align with the general trend and perform better than using compute measures like training FLOPs.

**Preregistered predictions on Open LLM leaderboard v2 benchmarks** Besides testing new models on exsiting benchmarks with preregistred predictions, we also tested observational scaling laws on the new, more challenging benchmarks being used in Open LLM Leaderboard v2. In particular, we selected a subset of new tasks where at least some exsiting open models demonstrate non-trivial performance (which will exclude benchmarks like IFEval and MUSR) and where scaling predictions from base benchmarks are non-trivial (which will exclude MMLU Pro), including GPQA [74], MATH [33], and BBH [83]. We fit both observational scaling laws and compute-based scaling laws on these benchmarks and compared their extrapolation performance. Since the tasks are more challenging, it requires a larger cutoff threshold to include more data points with non-trivial performance on these tasks. We set the FLOPs cutoff to be $16.8, 25.2, 8.4 \times 10^{21}$ for GPQA, MATH, and BBH, respectively. The results are in Fig. E.6. We find that observational scaling laws provides reasonable forecasts on these new, challenging benchmarks and outperform compute-based scaling laws when extrapolating to larger models.

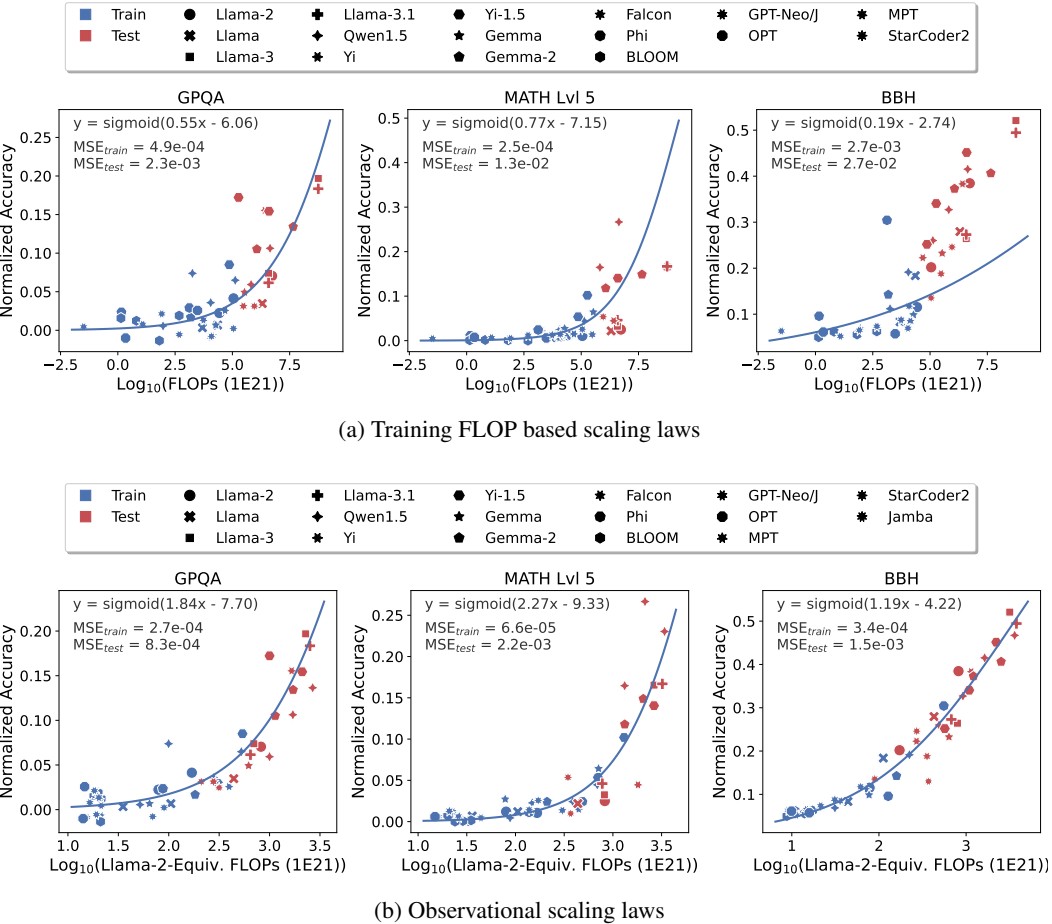

(a) Training FLOP based scaling laws

(b) Observational scaling laws

Figure E.6: Observational scaling laws also provide reasonable forcasts on new, more challenging benchmarks being used in Open LLM Leaderboard v2 and outperform compute-based scaling laws.

## E.4 Robustness Checks

**Number of PC selection**    Recall that we defaulted to use 3 PC measures for all of our prediction tasks. Here we provide additional analysis on the impact of using different numbers of PCs on the prediction performance and validate the robustness of our choice. In particular, we compare the fitted curves and prediction performance of using 1-4 PCs on all our tasks. The results are in Fig. E.7, Fig. E.8, and Fig. E.9 for post-training analysis, "emergent" capability, and agentic capability tasks, respectively. Our results indicate that using more than 2 PCs leads to better prediction performance than using compute measures like FLOPs, and using 3 PCs consistently leads to the most robust predictions across all the tasks. These validate our choice of using 3 PCs as the default number of PCs and indicate the robustness of our results to the choice of the number of PCs.

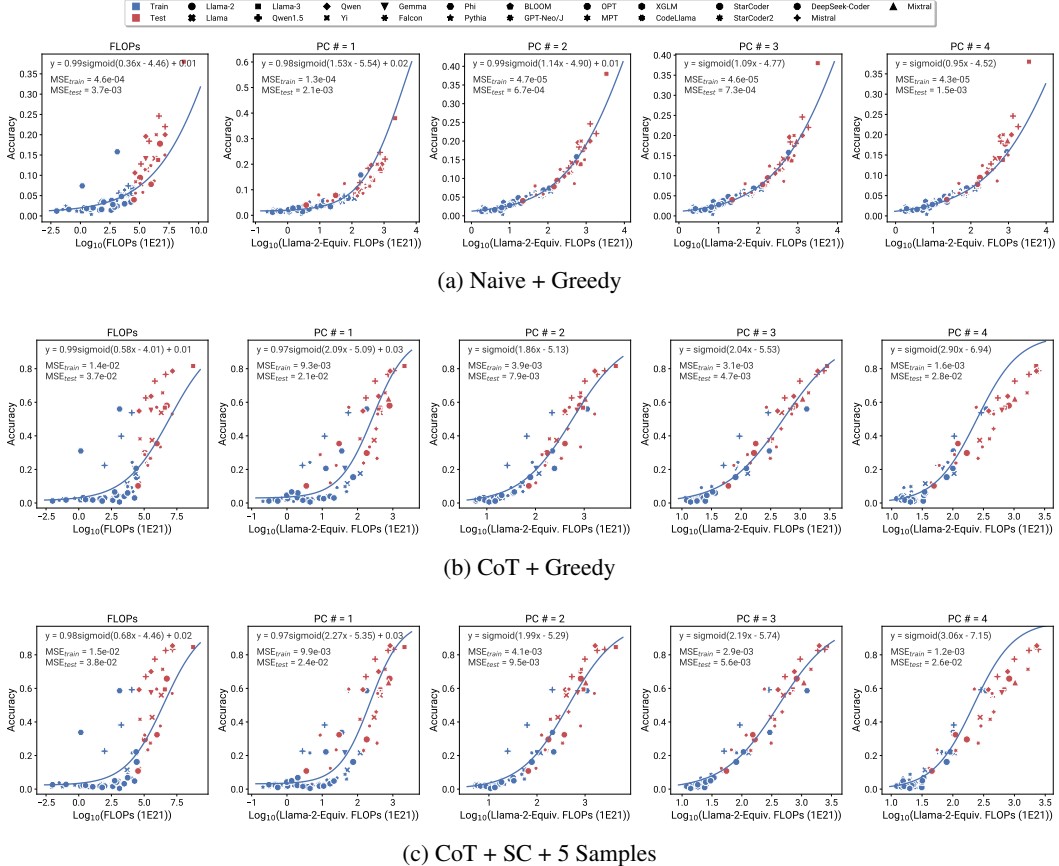

Figure E.7: Comparing the prediction performance of using different numbers of PCs for observational scaling laws on the **post-training analysis** tasks included in Sec. 4.3. Using PC measures consistently leads to better prediction performance than using compute measures like FLOPs with 3 PCs being the best across different tasks.

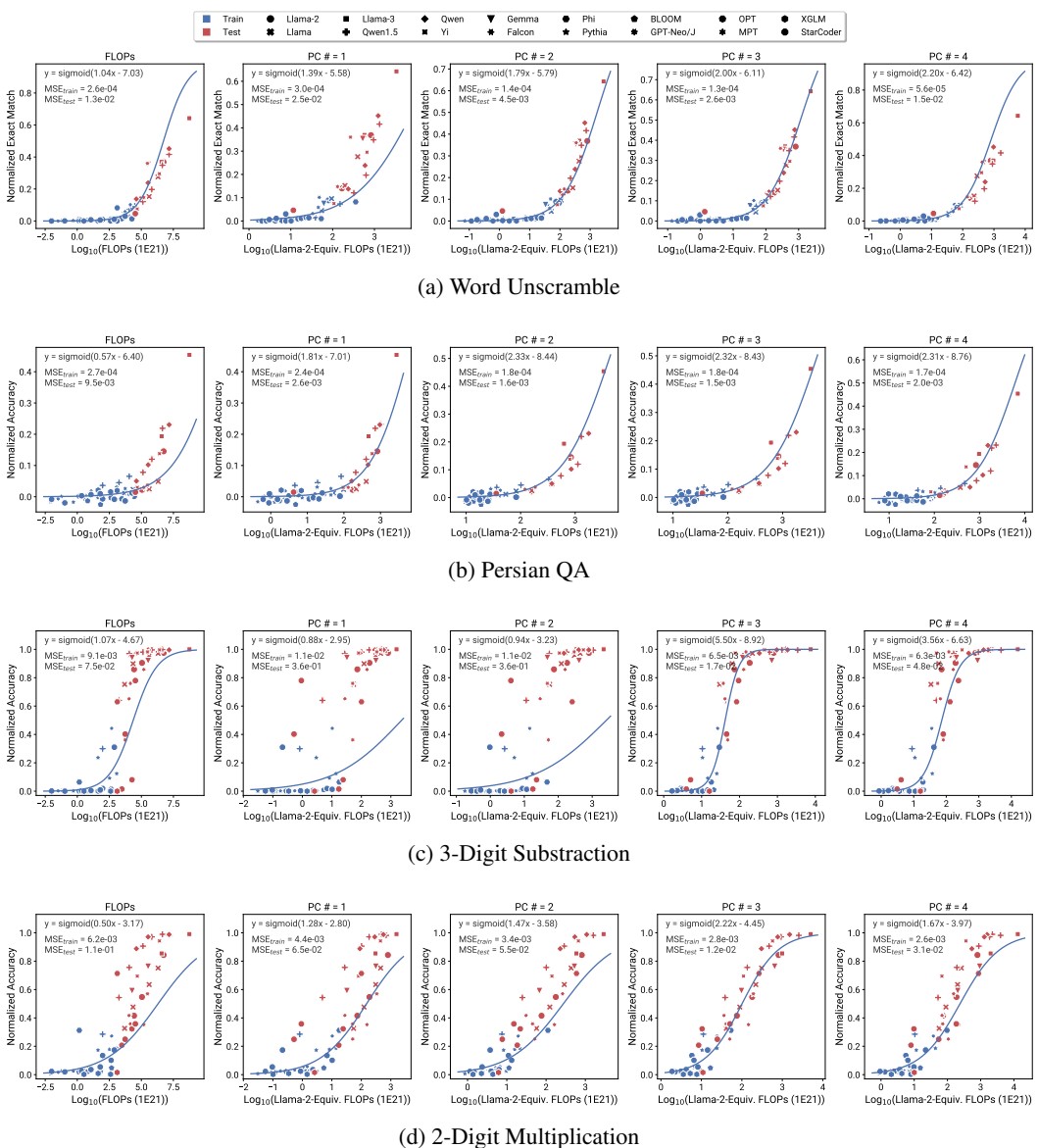

Figure E.8: Comparing the prediction performance of using different numbers of PCs for observational scaling laws on different **"emergent" capability** tasks included in Sec. 4.1. Using 3 PCs consistently leads to the best prediction performance across different tasks.

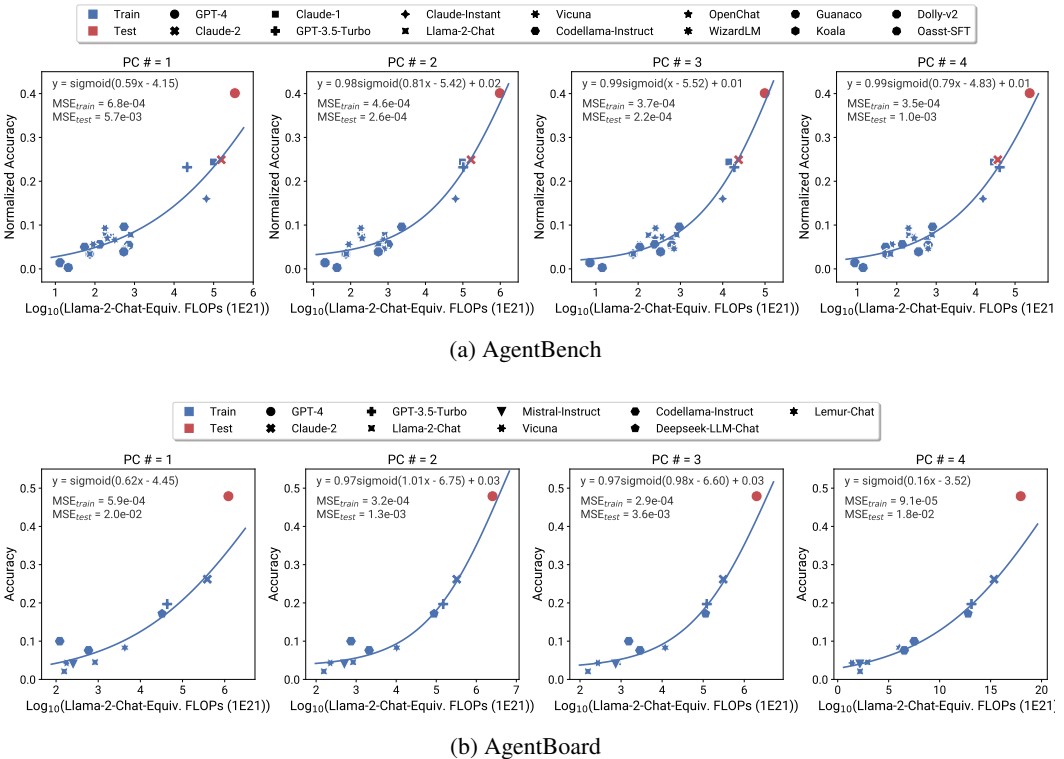

(a) AgentBench

(b) AgentBoard

Figure E.9: Comparing the prediction performance of using different numbers of PCs for observational scaling laws on the **agentic capability** tasks included in Sec. 4.2. Using 2 or 3 PCs leads to the best prediction performance across different tasks.

**Holdout cutoff selection**    The cutoff for selecting the holdout set could have a significant impact on the prediction performance of observational scaling laws, as it determines the size of the training set that could be crucial when the entire dataset is not large (as in our case). Here we analyze how the prediction performance changes with different holdout cutoffs for various predictive measures (PCs vs compute measures) and provide a quantitative comparison that characterizes their overall prediction performance under varying cutoffs.

Specifically, we conducted the analysis on the post-training analysis tasks in Sec. 4.3 and the "emergent" capability tasks in Sec. 4.1, where there are more data points (compared to the agentic capability tasks in Sec. 4.2) to provide a more robust analysis. For each task, we vary the FLOPs cutoff to control the ratio of the test set from 60% to 5% (linearly spaced), which consequently changes the difficulty of the prediction task from more difficult (less training data with weaker performance) to easier (more training data with stronger performance). We can then compare the test MSE of using different predictive measures under different cutoffs and quantify the overall prediction performance using the area under the error curve (AUE). For "emergent" capability tasks, we additionally include a variant of the cutoff strategy that holds out test data based on the accuracy on the task, which simulates a more challenging weak-to-strong prediction scenario and offers an extra robust analyses.

The results are depicted in Fig. E.10 and Fig. E.11. We observe that in most of our evaluated setups, using our PC measures (especially with 3 PCs) generally leads to an earlier transition to the low prediction error region and much lower AUE compared to using compute scales like training FLOPs and model size. This indicates that PC measures are more robust under different cutoffs and more sample-efficient for scaling analysis.

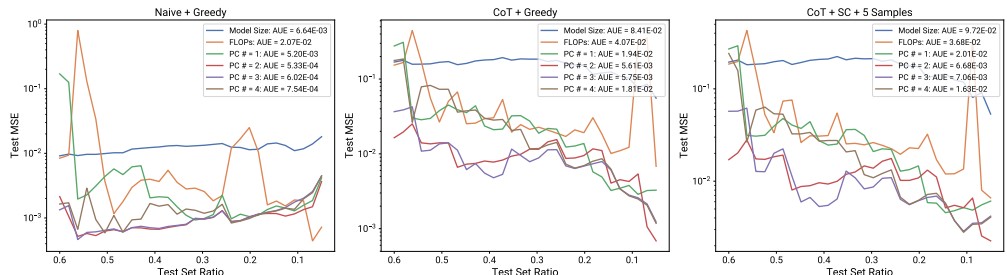

Figure E.10: Comparing different scale measures under different holdout cutoffs on post-training analysis tasks in Sec. 4.3. The training/test data size is varied by changing the FLOPs cutoff and the area under the test error curves (AUE) is used to measure the overall prediction errors. PC measures (with # = 2 or 3) consistently lead to an earlier transition to low prediction error region and much lower AUE compared to compute measures like training FLOPs and model size.

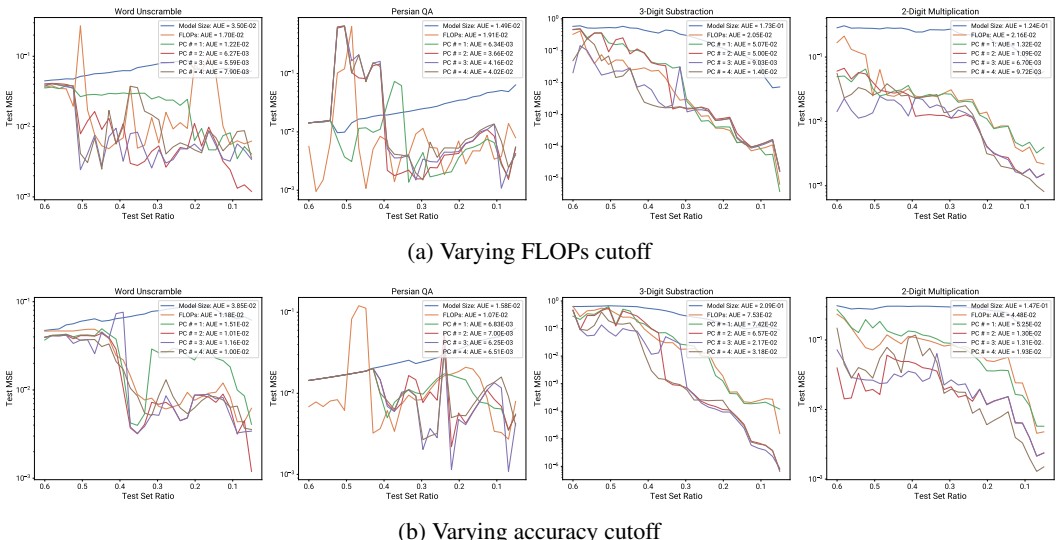

Figure E.11: Comparing different scale measures under different holdout cutoffs on "emergent" capability tasks in Sec. 4.1. The training/test data size is varied by changing the FLOPs (a) or accuracy (b) cutoff and the area under the test error curves (AUE) is used to measure the overall prediction errors. In 7 out of 8 setups, PC measures (with # = 3) lead to much lower AUE compared to compute measures like training FLOPs and model size.

### E.5 Emergent Capabilities

**Predicting with model sizes** In Fig. E.12, we show the prediction performance of using model size for the "emergent" capabilities of LMs. We find that it leads to significantly worse forecasts compared to using training FLOPs and PC measures and poorly captures the "emergence" trend. This is probably because models from different families were trained with very different data sizes and quality and may use different architectures.

**Using default cutoff for arithmetic tasks** In Fig. 4, we applied a different FLOPs cutoff than the default one on arithmetic tasks to make the prediction tasks more challenging. Here, we present the results of using the default FLOPs cutoff on arithmetic tasks in Fig. E.13. We find that using the default FLOPs cutoff makes the prediction tasks trivial with too many data points close to perfect performance. Notably, using PC measures still outperforms using compute measures like model size and training FLOPs, indicating its robustness to the choice of the cutoff.

**Additional tasks** In Fig. E.14, we present the results on additional "emergent" capability tasks included in Wei et al. [98]. Similar to the main tasks (Fig. 4), we used the default FLOPs cutoff for non-arithmetic tasks (IPA Transliterate) and a quarter of the default cutoff for arithmetic tasks (3-Digit Addition, 2-Digit Addition). We find that using PC measures consistently leads to the best prediction performance compared to using model size or training FLOPs. While the extrapolation does not exactly match the trend of the ground truth on the IPA Transliterate task, possibly due to the fact that the specific task capabilities are not well covered by our collected benchmark metrics, it still provides a reasonable forecast of the "emergence" behavior.

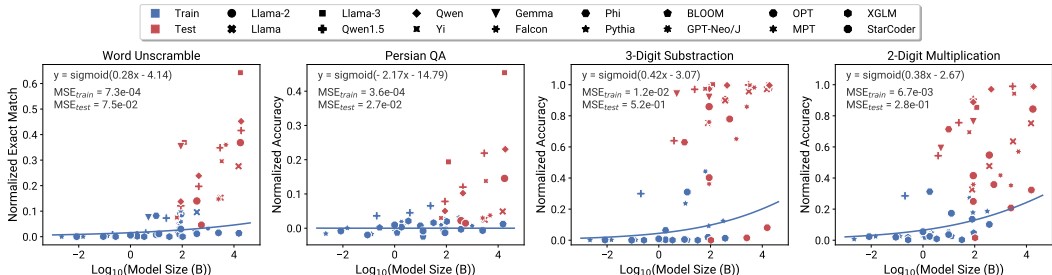

Figure E.12: Using model sizes gives poor predictions for the "emergent" capabilities of LMs.

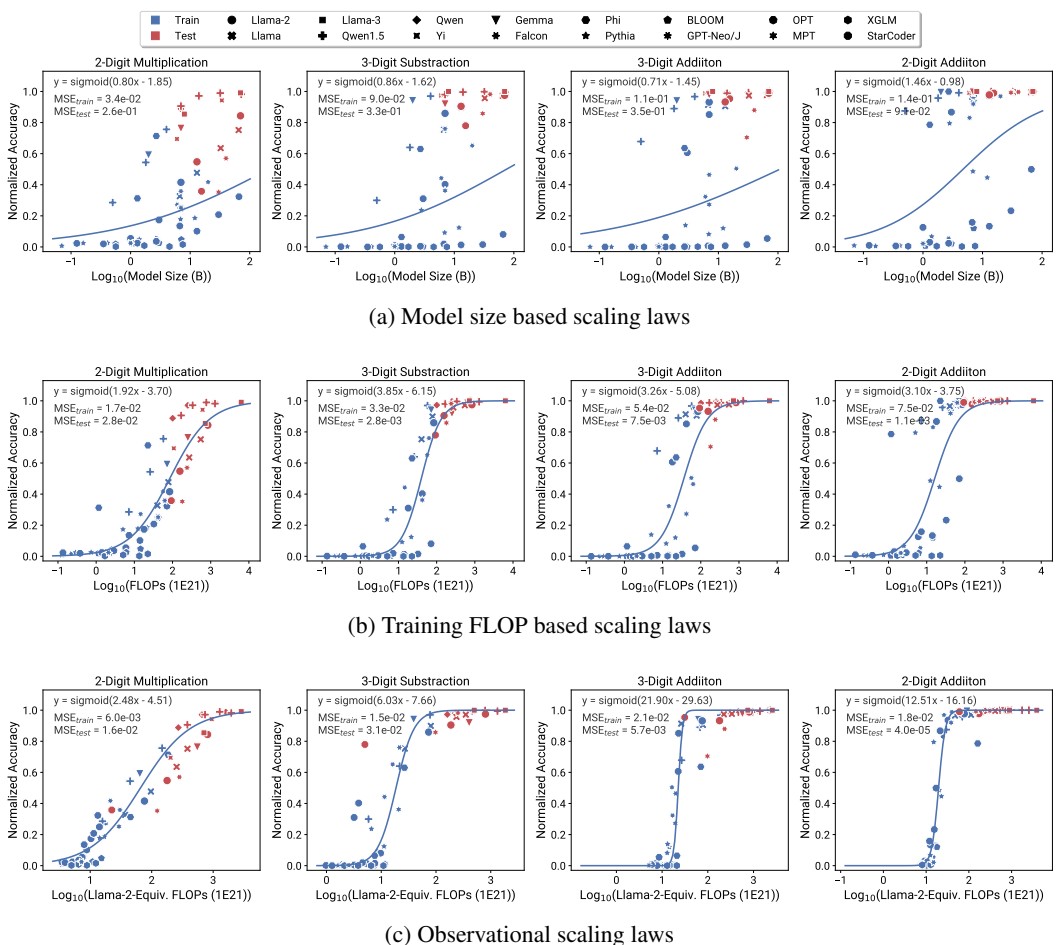

(a) Model size based scaling laws

(b) Training FLOP based scaling laws

(c) Observational scaling laws

Figure E.13: Using the default FLOPs cutoff on arithmetic tasks makes the prediction tasks trivial with too many data points close to perfect performance. Observational scaling laws using PC measures (with # = 3) still outperform compute scaling laws using model size and training FLOPs.

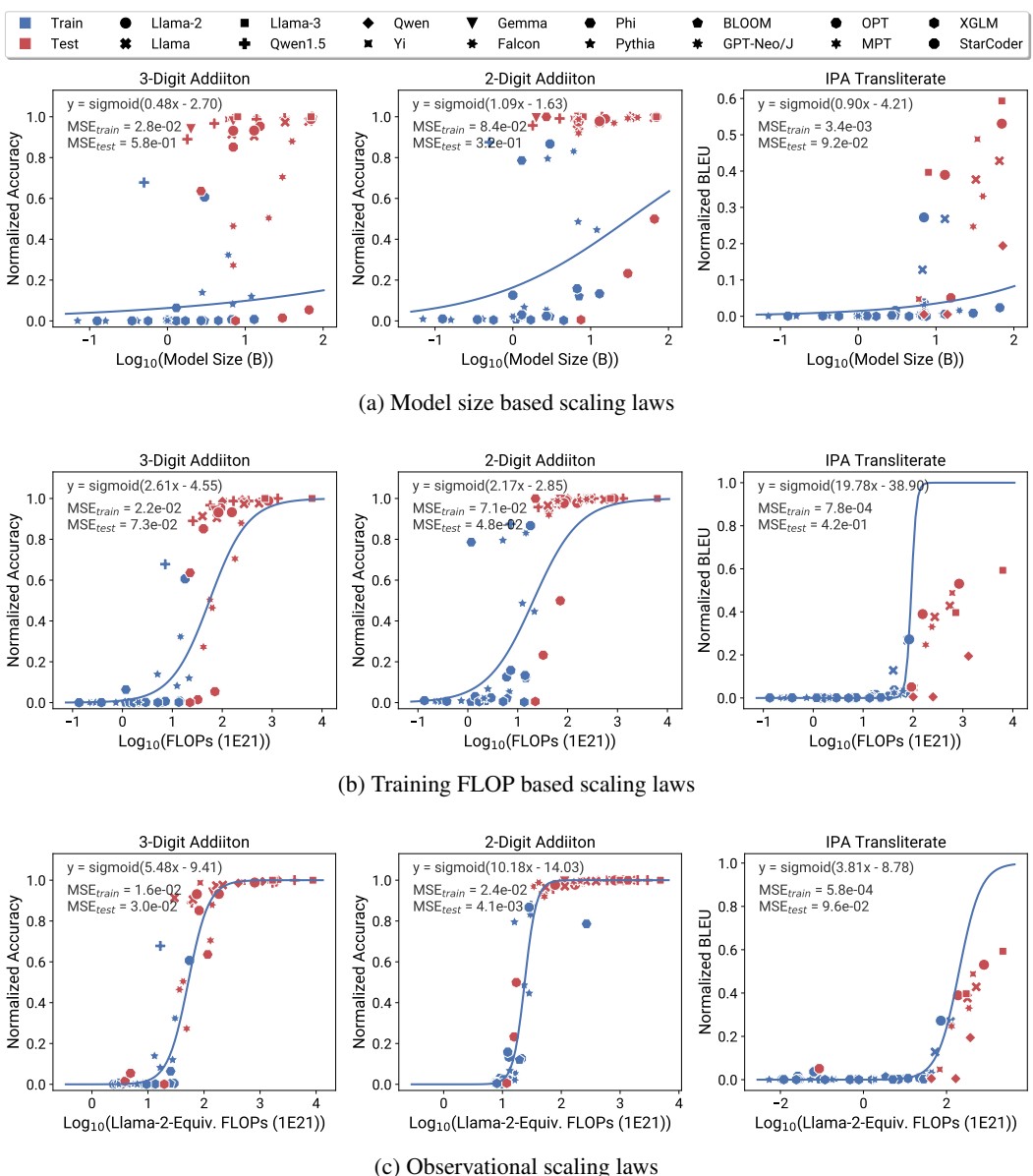

(a) Model size based scaling laws

(b) Training FLOP based scaling laws

(c) Observational scaling laws

Figure E.14: Results on additional "emergent" capability tasks included in Wei et al. [98]. Observational scaling laws using PC measures (with # = 3) consistently lead to the best prediction performance compared to compute scaling laws using model size and training FLOPs. Although the extrapolation does not exactly match the trend of the ground truth on the IPA Transliterate task, it still provides a reasonable forecast of the "emergence" behavior.

## E.6 Post-Training Method Analysis

**Prediction results with different scale measures** In Fig. E.15, we show the prediction performance of using different scale measures on various prediction tasks for the post-training method analysis on GSM8K. Similarly, using PC measures well captures the scaling trend and consistently leads to the best prediction performance across all tasks.

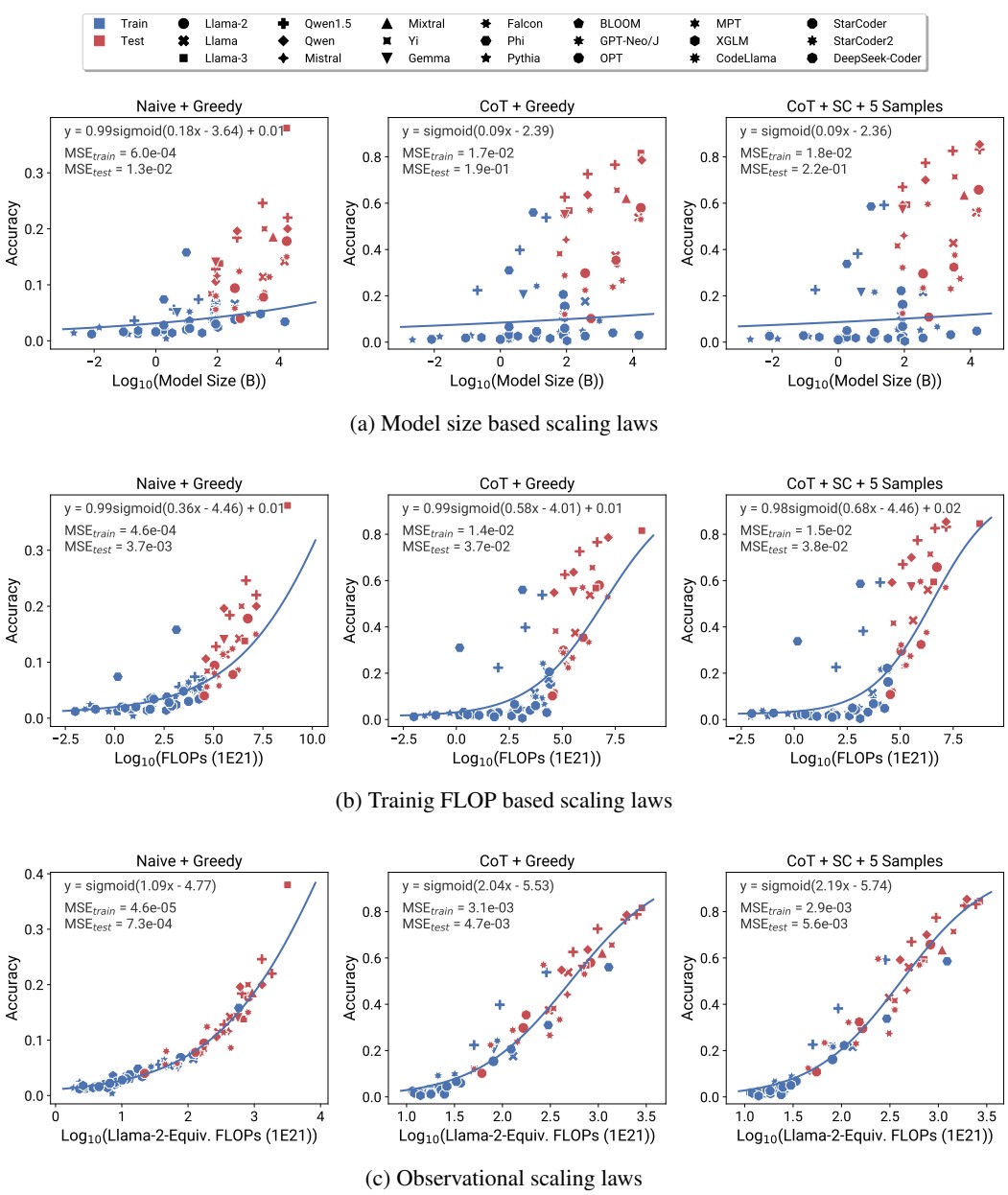

Figure E.15: Predicting the impact of post-training techniques on GSM8K with different scale measures. Observational scaling laws using PC measures (with # = 3) consistently lead to the best prediction performance across all tasks.

**Results on BBH** We further validated our observational scaling laws for predicting the impact of CoT on the BigBench-Hard tasks [83] following the same setup in Sec. 4.3. In particular, we used the defaulted FLOPs cutoff and the same PC measures (# = 3). We normalized the prediction accuracy on each BBH task by their respective random prediction accuracy and aggregated the normalized accuracy across all tasks for predictions. The results are depicted in Fig. E.16. Surprisingly, we observe that using training FLOPs leads to reasonable predictions of LM performance with and without CoT on BBH tasks, possibly due to the denoising effect of aggregation over all tasks. Furthermore, using PC measures accurately captures the scaling trends in both setups, even when using training FLOPs leads to less tight captures in the "Naive" setup or fails to capture the behavior of models trained on synthetic data (Phi).

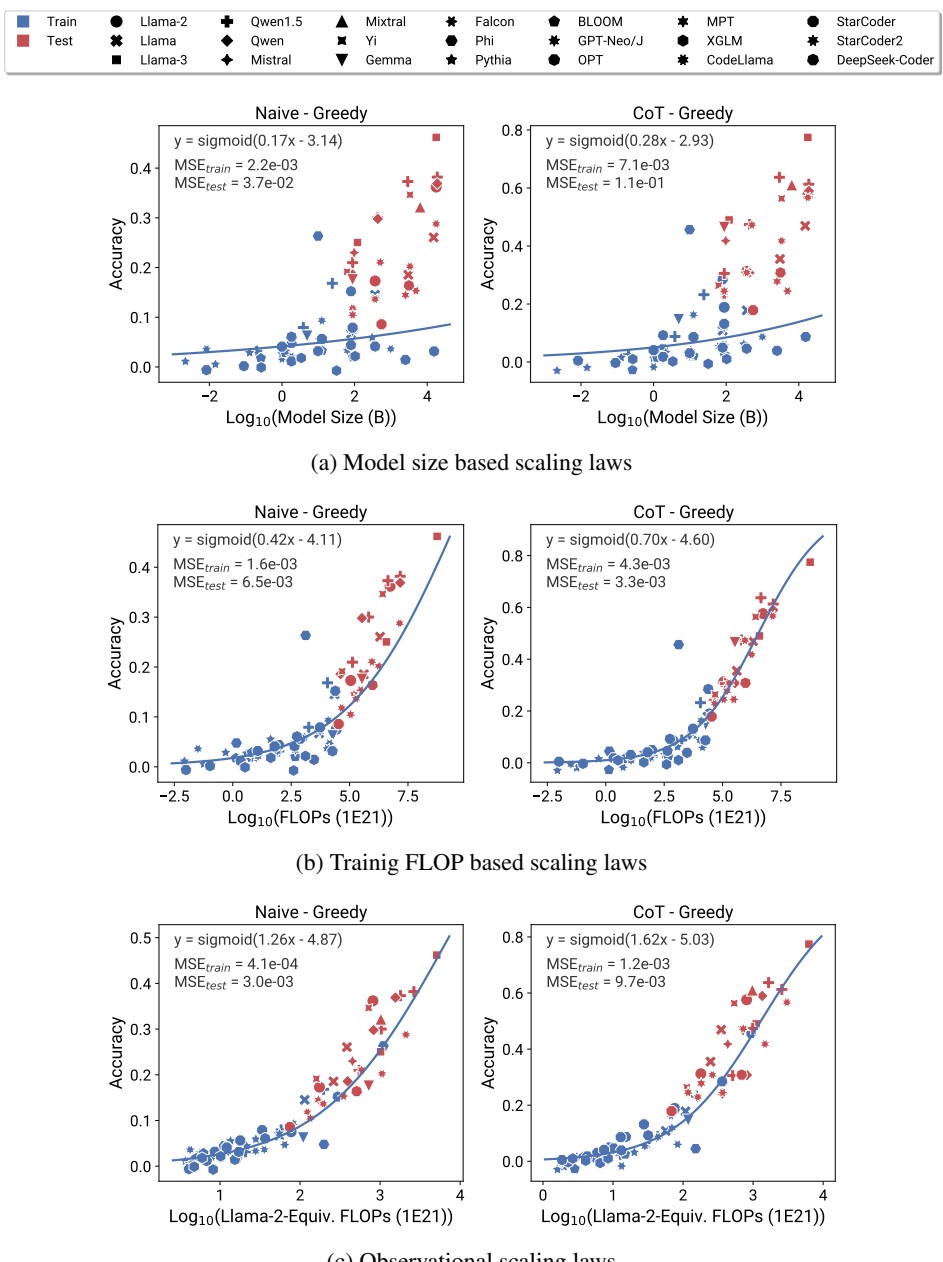

(a) Model size based scaling laws

(b) Trainig FLOP based scaling laws

(c) Observational scaling laws

Figure E.16: Predicting the impact of CoT on BBH tasks. Both using training FLOPs and PC measures leads to reasonable predictions, while PC measures accurately capture the scaling trends in both setups, even when using training FLOPs leads to less tight captures in the "Naive" setup or fails to capture the Phi model (which was trained on synthetic data) as an outlier.

### E.7 Model Subset Selection

**Prediction results with different number of models selected by V-optimality** In Fig. 7a, we demonstrated how the prediction errors change with the number of models selected by our method. Here we present a qualitative analysis of the prediction results with different numbers of models selected in Fig. E.17. We find that with more than 8 models, the fitted scaling curves have already converged to accurately capture the scaling trend, indicating the efficiency of our method.

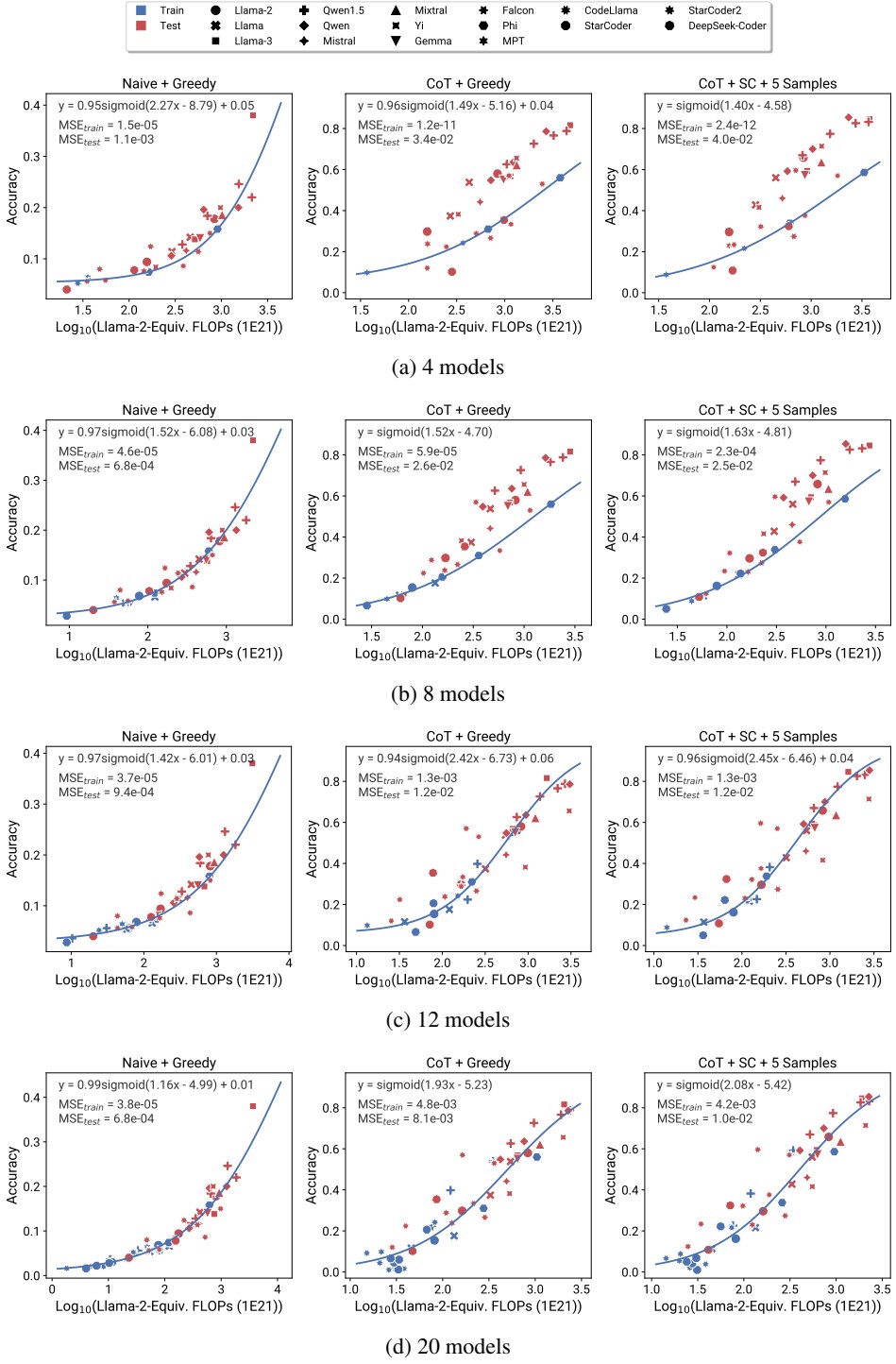

Figure E.17: Prediction results with different numbers of models selected with our V-optimality criterion. The predictions have accurately captured the scaling trend with more than 8 models.

**Prediction results with randomly selected models**  We present the prediction results with randomly selected models from all available models in Fig. E.18, in comparison to the results with models selected by our V-optimality criterion (Fig. E.17). All these results are produced with a fixed random seed. We find that using randomly selected models leads to a much worse prediction performance, even with 16 models, demonstrating the critical need to carefully select models for effective scaling analyses.

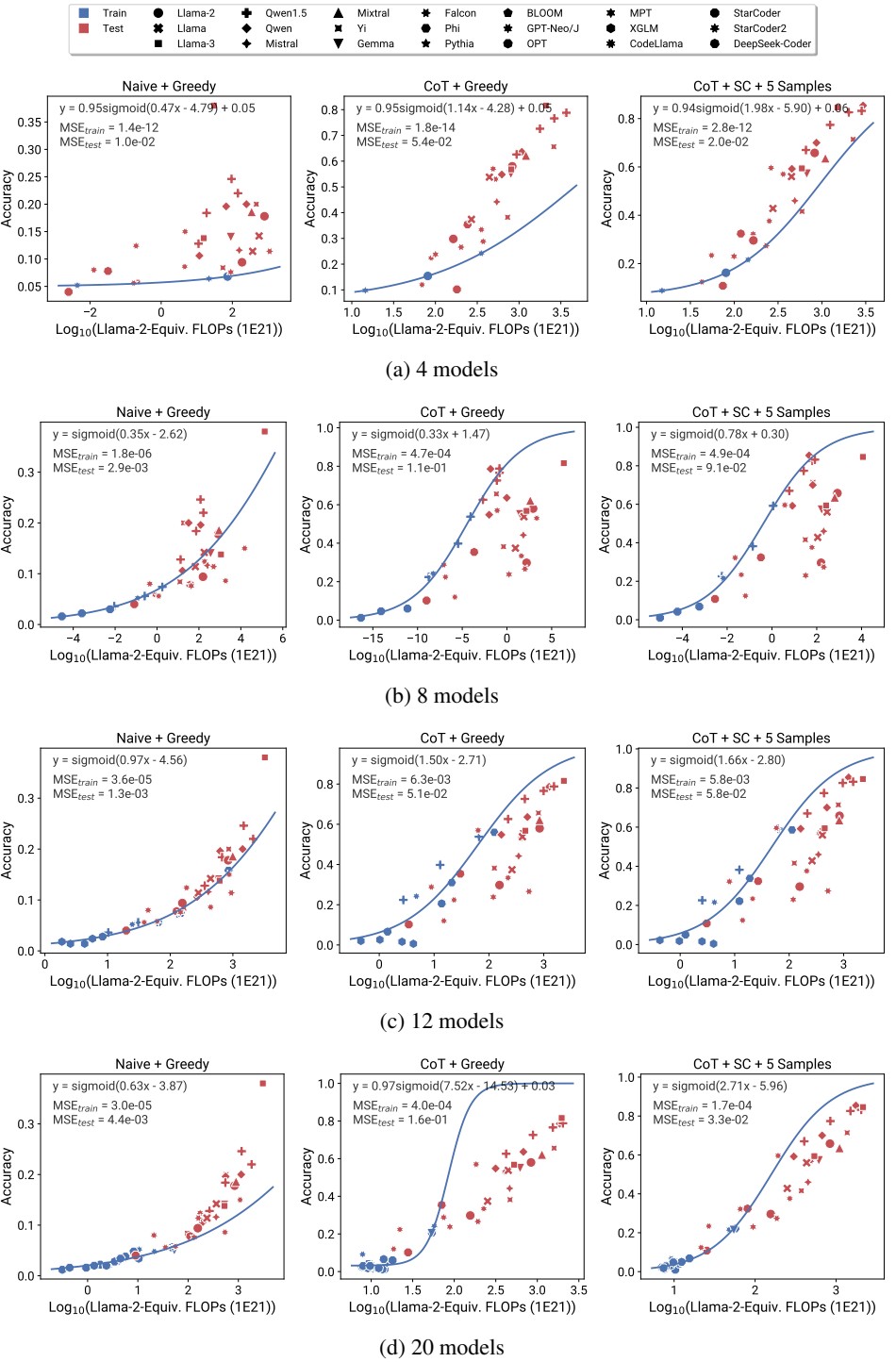

Figure E.18: Prediction results with different numbers of randomly selected models. The prediction performance is much worse than our selection method, even when 20 models are being selected.

Table E.1: Selected models for scaling analysis of post-training methods under different budgets.

| Budget | Selected Models |
|---|---|
| 8 models | Llama-2 {7B, 13B, 70B}, Mixtral {8x7B}, Phi {1.5B, 2}, MPT {7B, 30B} |
| 12 models | Llama-2 {7B, 13B, 70B}, Llama-3 {8B, 70B}, DeepSeek-Coder {1.3B, 6.7B, 33B}, Falcon {1B, 7B, 40B, 180B} |
| 20 models | Llama-2 {7B, 13B, 70B}, Mixtral {8x7B}, Qwen {7B, 14B, 72B}, DeepSeek-Coder {1.3B, 6.7B, 33B}, CodeLlama {7B, 13B, 34B, 70B}, MPT {7B, 30B}, Falcon {1B, 7B, 40B, 180B} |
| 8 models, sub 7B | Llama-2 {7B}, Llama {7B}, Qwen {7B}, DeepSeek-Coder {1.3B, 6.7B}, Phi {1.5, 2}, MPT {7B} |
| 12 models, sub 7B | Llama-2 {7B}, Llama {7B}, Qwen {7B}, DeepSeek-Coder {1.3B, 6.7B}, Phi {1.5, 2}, MPT {7B}, Gemma {2B, 7B}, Falcon {1B, 7B} |

**Recommended model series for scaling analysis**    To facilitate future scaling analyses for post-training techniques, we provide a reference list of models selected with our method under different budget constraints in Table E.1. These models were chosen from all available ones (see Table D.1) with Llama-2 models always being included (as it is currently the most representative and widely used model family), and are expected to be representative of them. Notably, the selected models cover diverse capability ranges and dimensions to capture potential scaling dimensions. For example, under the 12 model budget constraint, the selected models cover both stronger models (Llama-3) and weaker ones (Falcon), as well as models with specialized programming capabilities (DeepSeek-Coder). Updating this list with other constraints (e.g., total inference FLOPs) or new model families is straightforward, and we provide both implementations and guidelines in our released code.

### E.8 Additional Analysis

We have received valuable feedback from anonymous reviewers and have conducted extrnsive additional analysis to address their remaining questions.

**Extracting PC measures with non-matrix factorization**   We note that the benchmark coefficients on our principal capability measures are not guaranteed to be non-negative, which may hinder the interpretability of the extracted components. Therefore, we conduct an additional analysis with non-negative matrix factorization (NMF) to ensure the non-negativity of the component-wise benchmark coefficients that may provide more interpretable capability dimensions. The results are included in Fig. E.19. We observed the NMF components do generally demonstrate a interpretable decomposition, as well as a positive and smooth scaling with training FLOPs within each model family (as our PC measures).

While NMF offers enhanced interpretability and positive scaling properties compared to PCA, it also has notable limitations. Firstly, unlike PCA, NMF does not enforce orthogonality among its extracted components, as evident in the observed correlation between Components 3 and 4. Consequently, the coefficients assigned to each model across dimensions may not serve as independent measures of specific capabilities. Secondly, the ordering of NMF components lacks uniqueness and intrinsic physical meaning. This contrasts with PCA components, which are systematically ordered by their explained variances. The PCA approach provides an 'importance' measure for each dimension and allows for controlled trade-offs between representativeness and noise inclusion by adjusting the number of PCs used in the analysis.

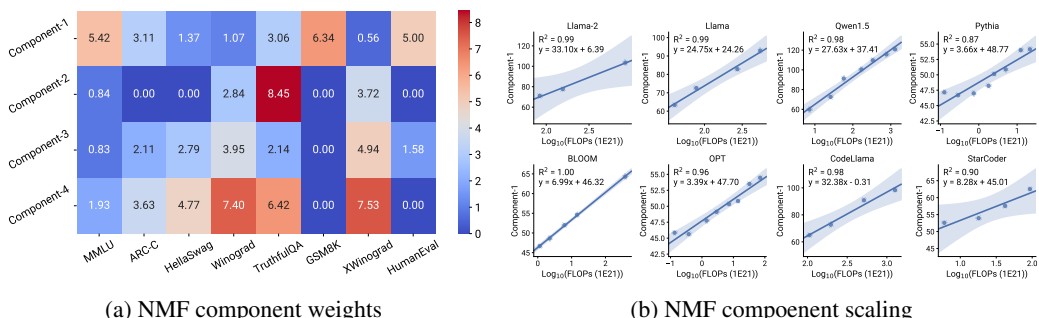

(a) NMF component weights    (b) NMF compoenent scaling

Figure E.19: **Extracting PC measures with non-matrix factorization**: More interpretable principal capability measures can be obtained by non-negative matrix factorization (NMF). (a) NMF ensures the non-negativity of the component-wise benchmark coefficients and provides an interpretable decomposition. For example, we may view component 1 and 4 as reasoning and language understanding capabilities, respectively. (b) The NMF components generally demonstrate a smooth, positive scaling with increasing FLOPs. The results also hold across other model families and components.

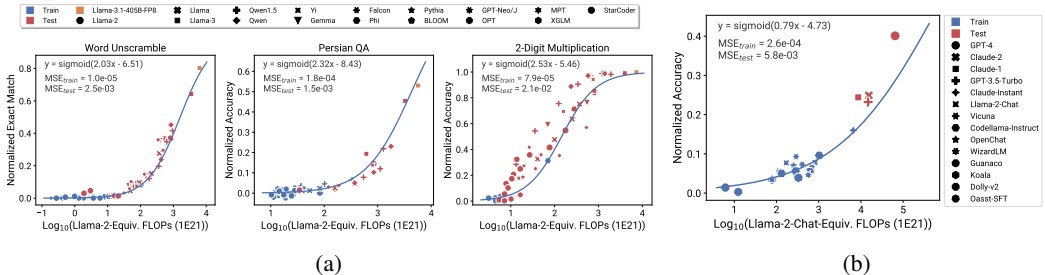

(a)                                                                (b)

Figure E.20: **Pushing the limit of cutoff point**: The cutoff can be further pushed back on each individual task while still providing reasonable predictions. (a) Emergent capability tasks: We include three representative tasks, and the task-specific FLOPs cutoff are 25, 84, and $8 \times 10^{21}$ respectively (from left to right), compared to the unified $84 \times 10^{21}$ in our current setup. We also test the newly released Llama-3.1 405B (FP8) to assess the generalization to a larger scale. (b) Agentic tasks: We test on AgentBench that has more available data points with an 80/20 train/test split. The extrapolations underestimate performance to some extent, but still align with the overall observed trend.

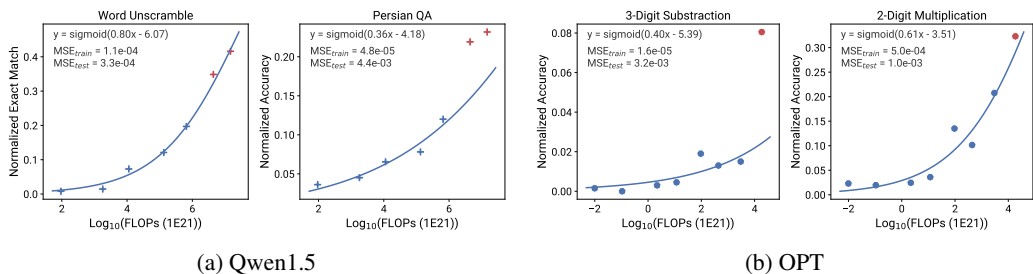

(a) Qwen1.5                                                    (b) OPT

Figure E.21: **Scaling predictions with single-family models**: For scaling prediction from FLOPs within a single family, at least 5 models are typically required for accurate extrapolation, but the performance is highly dependent on the specific setup. We test Qwen1.5 on non-algorithmic and OPT on arithmetic tasks. Both model families demonstrate accurate extrapolation on one task but not the other.

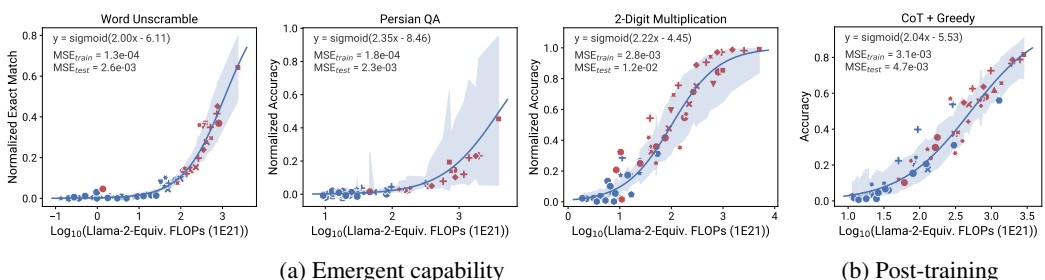

(a) Emergent capability                                    (b) Post-training

Figure E.22: **Confidence intervals of scaling predictions**: We calculate 95% confidence intervals for predictions from the non-linear regression models at each data point, and the observed data points fall within these confidence intervals. When extrapolating from very few data points above the random-guess level (e.g., in Persian QA), the confidence intervals may be wider. We include representative tasks for both emergent capability (left) and post-training analysis (right) setups.

### E.9 Fitted Functional Forms for Preregistration of Predictions

In Table E.2, we included the functional forms of fitted scaling laws in our experiments. These functional forms served as a preregistration of our predictions for future models at the time of the initial paper release, which has been used to test the generalizability of our scaling analysis to unseen models.

Table E.2: The functional forms of the fitted scaling laws included in our paper, are preregistered for predictions of future models. Each functional form is presented as the logit of the normalized accuracy metric $\phi^{-1}(Y, h) = \sigma^{-1}\left((Y - (1 - h))/h\right) = X$ that is equivalent to Eq. (6). Each benchmark metric is scaled to be within the range $[0, 1]$.

| Setup | Task | Functional Form |
|---|---|---|
| **"Emergent" capabilities** (Sec. 4.1) | Word Unscramble | $\phi^{-1}(Y, 1.00)$ 
 $= 2.00\log_{10}(\bar{C}_{\text{Llama-2}}) - 6.11$ 
 $= 6.74\text{PC1} - 3.22\text{PC2} - 1.37\text{PC3} - 4.93$ 
 $= 1.02\text{MMLU} + 3.02\text{ARC-C} + 5.73\text{HellaSwag} + 2.44\text{Winograd} -$ 
 $1.06\text{TruthfulQA} + 1.21\text{GSM8K} + 2.48\text{XWinograd} - 0.08\text{HumanEval} - 12.28$ |
| | Persian QA | $\phi^{-1}(Y, 1.00)$ 
 $= 2.32\log_{10}(\bar{C}_{\text{Llama-2}}) - 8.43$ 
 $= 2.86\text{PC1} + 3.18\text{PC2} - 0.19\text{PC3} - 5.26$ 
 $= 2.08\text{MMLU} + 1.06\text{ARC-C} + 1.13\text{HellaSwag} + 0.53\text{Winograd} +$ 
 $0.36\text{TruthfulQA} + 2.89\text{GSM8K} + 0.66\text{XWinograd} + 1.55\text{HumanEval} - 7.98$ |
| | 3-Digit Substraction | $\phi^{-1}(Y, 1.00)$ 
 $= 5.50\log_{10}(\bar{C}_{\text{Llama-2}}) - 8.92$ 
 $= 5.98\text{PC1} + 8.74\text{PC2} + 39.55\text{PC3} - 4.68$ 
 $= 2.17\text{MMLU} + 2.32\text{ARC-C} - 3.44\text{HellaSwag} - 7.96\text{Winograd} +$ 
 $0.65\text{TruthfulQA} + 34.27\text{XWinograd} + 20.39\text{HumanEval} - 20.99$ |
| | 2-Digit Multiplication | $\phi^{-1}(Y, 1.00)$ 
 $= 2.22\log_{10}(\bar{C}_{\text{Llama-2}}) - 4.45$ 
 $= 3.60\text{PC1} + 4.24\text{PC2} + 8.05\text{PC3} - 2.68$ 
 $= 1.62\text{MMLU} + 1.95\text{ARC-C} + 0.55\text{HellaSwag} - 0.63\text{Winograd} +$ 
 $0.14\text{TruthfulQA} + 6.80\text{XWinograd} + 6.52\text{HumanEval} - 8.00$ |
| **Agentic capabilities** (Sec. 4.2) | AgentBench | $\phi^{-1}(Y, 0.99)$ 
 $= \log_{10}(\bar{C}_{\text{Llama-2-Chat}}) - 5.52$ 
 $= 2.32\text{PC1} + 0.79\text{PC2} - 2.82\text{PC3} - 2.96$ 
 $= 2.34\text{MMLU} + 0.82\text{ARC-C} + 0.32\text{HellaSwag} + 0.54\text{Winogrande} +$ 
 $0.60\text{TruthfulQA} - 0.42\text{GSM8K} + 2.63\text{HumanEval} - 6.37$ |
| | AgentBoard | $\phi^{-1}(Y, 0.97)$ 
 $= 0.98\log_{10}(\bar{C}_{\text{Llama-2-Chat}}) - 6.60$ 
 $= 3.02\text{PC1} + 2.60\text{PC2} + 1.17\text{PC3} - 2.98$ 
 $= -0.10\text{MMLU} - 0.31\text{ARC-C} - 0.55\text{HellaSwag} + 0.14\text{Winogrande} +$ 
 $0.56\text{TruthfulQA} + 2.28\text{GSM8K} + 3.36\text{HumanEval} - 5.06$ |

| Setup | Task | Functional Form |
|---|---|---|
| **Post-training** (analysis Sec. 4.3) | GSM Naive + Greedy | $\phi^{-1}(Y, 1.00)$ 
 $= 1.09 \log_{10}(\bar{C}_{\text{Llama-2}}) - 4.77$ 
 $= 2.69\text{PC1} + 1.55\text{PC2} - 0.36\text{PC3} - 3.57$ 
 $= 1.53\text{MMLU} + 1.30\text{ARC-C} + 1.22\text{HellaSwag} + 0.75\text{Winograd} +$ 
 $0.16\text{TruthfulQA} + 0.13\text{XWinograd} + 1.92\text{HumanEval} - 5.97$ |
| | GSM CoT + Greedy | $\phi^{-1}(Y, 1.00)$ 
 $= 2.04 \log_{10}(\bar{C}_{\text{Llama-2}}) - 5.53$ 
 $= 2.56\text{PC1} + 4.64\text{PC2} + 4.21\text{PC3} - 2.50$ 
 $= 5.03\text{MMLU} + 2.04\text{ARC-C} - 0.10\text{HellaSwag} + 0.96\text{Winograd} +$ 
 $1.75\text{TruthfulQA} - 2.39\text{XWinograd} + 2.58\text{HumanEval} - 4.77$ |
| | GSM CoT + SC | $\phi^{-1}(Y, 1.00)$ 
 $= 2.19 \log_{10}(\bar{C}_{\text{Llama-2}}) - 5.74$ 
 $= 2.73\text{PC1} + 4.82\text{PC2} + 4.95\text{PC3} - 2.49$ 
 $= 5.58\text{MMLU} + 2.27\text{ARC-C} - 0.08\text{HellaSwag} + 1.11\text{Winograd} +$ 
 $1.97\text{TruthfulQA} - 2.78\text{XWinograd} + 2.45\text{HumanEval} - 4.95$ |
| | BBH Naive + Greedy | $\phi^{-1}(Y, 1.00)$ 
 $= 1.26 \log_{10}(\bar{C}_{\text{Llama-2}}) - 4.87$ 
 $= 2.70\text{PC1} + 3.06\text{PC2} - 0.84\text{PC3} - 3.23$ 
 $= 1.41\text{MMLU} + 1.05\text{ARC-C} + 0.75\text{HellaSwag} + 0.36\text{Winograd} +$ 
 $0.11\text{TruthfulQA} + 0.61\text{XWinograd} + 3.63\text{HumanEval} - 5.47$ |
| | BBH CoT + Greedy | $\phi^{-1}(Y, 1.00)$ 
 $= 1.62 \log_{10}(\bar{C}_{\text{Llama-2}}) - 5.03$ 
 $= 4.20\text{PC1} + 3.81\text{PC2} - 2.92\text{PC3} - 3.12$ 
 $= 0.84\text{MMLU} + 1.30\text{ARC-C} + 1.57\text{HellaSwag} + 0.42\text{Winograd} -$ 
 $0.44\text{TruthfulQA} + 1.96\text{XWinograd} + 5.62\text{HumanEval} - 6.61$ |

