# OpenReview forum: "Observational Scaling Laws and the Predictability of Langauge Model Performance"
_NeurIPS.cc/2024/Conference — NeurIPS 2024 spotlight_

### Official Review · Reviewer_DfgG · 2024-07-09

**Soundness:** 3
**Presentation:** 4
**Contribution:** 4
**Rating:** 7
**Confidence:** 4

**Summary:**

This paper proposes observational scaling laws to align scaling laws of computing from different model families, which are trained on various recipes, by projecting model benchmark performance to surrogate compute. This enables applying scaling law analysis without actually training models. Using the observational scaling laws, the authors predict emergent abilities, including agent performance of models trained with much more compute.

**Strengths:**

1. This paper presents an elegant approach to resolve the misalignment of scaling laws due to different training recipes (e.g., pretraining data).
2. This paper also shows the surrogate compute drawn from the proposed observational scaling laws can predict emergent abilities with sigmoid function.

**Weaknesses:**

1. The predictability of emergent abilities seems to be overclaimed. The provided results in Fig. 4, 6 and 7 support that the sigmoid function can well fit the relationship between downstream performance and (surrogate) compute. However, the feasibility of making reliable predictions on emergent ability is doubtful. The fitting data in Fig. 4  include points that have shown non-trivial performance and some even reached the inflection point.  And extrapolation results in Fig. 5 only scale 4x.

2. Projecting the model benchmark performance to low dimensional surrogates of compute seems to imply the assumption that different model families mainly differ in the speed to obtain abilities and have little difference in the trade-off of different capabilities. Let's say A models are trained with 1% code + 99% text, while B are trained with 99% code + 1% text, do observational scaling laws still hold for these two model families?

**Questions:**

1. Eqn. 3 and 4 indicate that model performance scale with log compute in sigmoids. But Eqn. 5 shows linear projection between model performance and surrogates of log compute. This misalignment seems weird.
2. What are the implications of observational scaling laws on evaluating pretraining data quality? Do different families of models show very different compute efficiency according to your results in aligning them?

**Limitations:**

The applicability of observational scaling laws seems to depend on training data distribution. The influence has not been discussed.

---

> ### Author Rebuttal · Authors · 2024-08-06
>
> Thank you for your valuable feedback and suggestions. We would like to address your remaining questions and concerns with the following responses.
>
>
> ### Capability predictability and cutoff selection
>
>
> > *“The predictability of emergent abilities seems to be overclaimed… However, the feasibility of making reliable predictions on emergent ability is doubtful. The fitting data in Fig. 4 include points that have shown non-trivial performance and some even reached the inflection point”*
>
> We would like to clarify that for the emergent capability setups, some data points above the random-guess region are needed since it is not possible to fit reliable scaling with only random-guess data points that extrapolate to the effective scaling region. Our cutoff point was carefully selected to unify all the setups as much as possible to avoid the concerns of "cutoff overfitting" to each specific setup, so there may be more non-trivial data points in some setups than others. We will adjust the claims and explain this point more explicitly in the paper.
>
> In addition, we have also included extra results of pushing back the cutoff point for each specific emergent capability task, included in Figure 1(a) in the supplementary PDF. We presented the results on three representative tasks due to space constraints and tested on the newly released **Llama-3.1 405B (FP8)** to assess the generalization to a larger scale. We observe that with a more aggressive cutoff setup, observational scaling laws can still provide reasonable predictions, even to a larger scale (Llama-3.1 405B) that was not available for testing before. We will include these additional results in our future version of the paper.
>
>
>
> > *”And extrapolation results in Fig. 5 only scale 4x.”*
>
> We would like to clarify that the X-axis is on the log-e instead of log-2 scale, so the extrapolation is about ~7.5. We commit to testing our current scaling predictions on more capable models once they are released, similar to what we have done in the emergent capability setups by adding the newly released Llama-3.1 405B (Figure 1(a) in the supplementary PDF), which will enable us to further check how much our predictions can scale.
>
>
> ### Applicability to different model families & data distribution
>
> > *”Projecting the model benchmark performance to low dimensional surrogates of compute seems to imply the assumption that different model families mainly differ in the speed to obtain abilities and have little difference in the trade-off of different capabilities. Let's say A models are trained with 1% code + 99% text, while B are trained with 99% code + 1% text, do observational scaling laws still hold for these two model families?”*
>
> The training data that different models are trained on will mainly determine their speed in converting to training compute to different capabilities (as shown in Figure 3), but not the observational scaling results. In our experiments, we have included models trained on mostly text (e.g., Llama), mostly code (StarCoder), multilingual data (BLOOM), and synthetic data (e.g., Phi), and observed consistent scaling predictions for these models.
>
>
>
>
> > *”What are the implications of observational scaling laws on evaluating pretraining data quality? Do different families of models show very different compute efficiency according to your results in aligning them?”*
>
> As discussed above, the training data distributions will determine the efficiency in converting the compute to different capabilities. For example, the models trained on pure code data may demonstrate better compute scaling for the PC component corresponding to the programming capabilities than models trained mostly on natural language. In our results, different model families indeed demonstrate different compute efficiencies in converting to different PC components, as shown by the varying slopes of different scaling curves in Figure 3.
>
>
>
> > *”The applicability of observational scaling laws seems to depend on training data distribution. The influence has not been discussed.”*
>
> As discussed above, we would like to clarify that the scaling predictions from observational scaling laws could accommodate models trained from different data training distributions, instead of “depending on” them, and our results have already accommodated many models trained from very different data distributions. The data distributions determine their compute efficiency in converting FLOPs to capabilities instead of from principal capabilities to more complex downstream performance (observational scaling).
>
>
>
>
> ### Formulation misalignment
>
> > *”Eqn. 3 and 4 indicate that model performance scale with log compute in sigmoids. But Eqn. 5 shows linear projection between model performance and surrogates of log compute. This misalignment seems weird.”*
>
> We could unify the two formulations, e.g., by adding a sigmoid relation in Eqn. 5, and it will probably also work reasonably well if we properly deal with the logit transformations during the floor (~0) and ceiling (~1) region. We choose the current formulation because it simply works well in practice and does not involve dealing with the potential ill-behaved, numerical issues of the logit transformation.

---

> > ### Comment · Reviewer_DfgG · 2024-08-12
> >
> > Thank you so much for the detailed reply.
> >
> > It is impressive to see the scaling trend holds for LLaMa 3.1 405B although I think the difference in training data has not been sufficiently resolved. The sigmoid curve of different model families do not overlap.
> >
> > Thank you so much for your inspiring work. Since the original rating is high, I will maintain my score.

---

### Official Review · Reviewer_oXqB · 2024-07-11

**Soundness:** 3
**Presentation:** 3
**Contribution:** 3
**Rating:** 6
**Confidence:** 3

**Summary:**

They propose using PCA decomposition of the performance of a range of models across a number of benchmarks to form an observational scaling law, which effectively predicts downstream performance across several different model families, including predicting post-training interventions like chain-of-thought. They find that 3 principal components explain much of the variance on the LM benchmarks that they test. They can then use these principal components to 1) derive a llama2 effective compute quantity to use as a universal x-axis that works across all model families; and 2) for a new task and model family fit an observational scaling law based on the relevant principal components. They demonstrate that they are able to accurately extrapolate the performance of more capable models using weaker models on a number of standard benchmarks, including emergent tasks from Big-Bench. They also show that they can predict GPT-4's agent capabilities, and the effect of techniques like chain of thought and self-consistency using their method. Finally they demonstrate an approach for selecting a minimal sufficient set of models for deriving the observational scaling law.

**Strengths:**

* Developing a better understanding of the scaling of downstream capabilities in language models is an important problem and their paper represents very promising progress towards this goal.
* Their experiments are pretty thorough, encompassing a range of models, tasks, and inference-time techniques.

**Weaknesses:**

* They show that their method can extrapolate predictions of stronger model capabilities using weaker models; however we don't get to see how far in advance they are able to make predictions. It would bo nice to see an ablation where the cutoff point used for fitting is pulled back so as to understand when extrapolation begins to fail. Moreover, they claim that they can predict emergence capabilities, however their cutoff point is conveniently just a little bit after performance begins to improve beyond chance, enabling their extrapolations to work. In the case where the cutoff point is pulled back further, are such emergent predictions still possible? I feel that demonstrating this would be important for being able to make the claim that prediction of emergence tasks is possible.
* They don't give any kind of uncertainty measure or confidence interval for their predictions.

**Questions:**

For a given take, are all models evaluated with the same prompts? It says you got some evals from Open LLM Leaderboard and others from EvalPlus and yet others from OpenLMLeaderboard. Are we sure all of these sources used the same evaluation procedure?

**Limitations:**

I think they mostly did a good job of acknowledging limitations.

---

> ### Author Rebuttal · Authors · 2024-08-06
>
> Thank you for your valuable feedback and suggestions. We appreciate that you acknowledged that our work represents “very promising progress” towards an important problem with thorough experiments. We would like to address your remaining concerns with the following responses.
>
>
> ### Ablation study on the cutoff point
>
> > *”It would bo nice to see an ablation where the cutoff point used for fitting is pulled back so as to understand when extrapolation begins to fail. Moreover, they claim that they can predict emergence capabilities, however their cutoff point is conveniently just a little bit after performance begins to improve beyond chance, enabling their extrapolations to work. In the case where the cutoff point is pulled back further, are such emergent predictions still possible?”*
>
> We would like to clarify that our cutoff point was carefully selected to unify all the setups as much as possible to avoid the concerns of "cutoff overfitting" to each specific setup. We have also made the unified cutoff point as hard as we could by sweeping over the cutoffs (see Figure E.9), where we found some signals from data points above the random-guess regions are needed for reliable scaling predictions, especially for emergent capability setups – it is not possible to fit reliable scaling with only random-guess data points that extrapolate to the effective scaling region. We will explain this more explicitly in the main text.
>
> For the ablation study on the cutoff point, we have already included a quantitative analysis in Figure E.9 where we measure the test error with varying cutoffs to control the training/test size. In addition, we have also included extra results of pushing back the cutoff point for each specific emergent capability task, included in Figure 1(a) in the supplementary PDF. We presented the results on three representative tasks due to space constraints and tested on the newly released **Llama-3.1 405B (FP8)** to assess the generalization to a larger scale. We observe that with a more aggressive cutoff setup, observational scaling laws can still provide reasonable predictions, even to a larger scale (Llama-3.1 405B) that was not available for testing before. We will include these additional results in our future version of the paper.
>
>
> ### Confidence intervals
>
> > *”They don't give any kind of uncertainty measure or confidence interval for their predictions.”*
>
> We have calculated the 95% confidence intervals for predictions from the non-linear regression models at each data point using parametric bootstrap. The results are included in Figure 4 in the supplementary PDF. We find that the observed data points mostly fall within these confidence intervals, though the intervals may be wide when there are very few effective data points above the random-guess region (e.g., in Persian QA).
>
>
>
>
>
> ### Unification of setups
>
> > *”For a given take, are all models evaluated with the same prompts? It says you got some evals from Open LLM Leaderboard and others from EvalPlus and yet others from OpenLMLeaderboard. Are we sure all of these sources used the same evaluation procedure?”*
>
> Yes, we have carefully checked the evaluation setups of these results to follow the same evaluation protocols (including prompts, few-shot examples, etc). For example, in the [documentation of Open LLM Leaderboard](https://huggingface.co/docs/leaderboards/open_llm_leaderboard/about#reproducibility), they have detailed how to conduct the evaluation following the same protocol. For the models we have evaluated by ourselves, we have also carefully followed the same procedure.

---

> > ### Comment · Reviewer_oXqB · 2024-08-13
> >
> > Thank you for your response. I appreciate the work to include comparisons of the cutoff point and also confidence intervals (although I don't seem to see this supplementary pdf on openreview that you are referring to). I'm willing to raise my score 1 point to a 6.

---

> > > ### Author Response · Authors · 2024-08-13
> > >
> > > Thank you very much for your reply. The supplementary pdf is included in the [global response](https://openreview.net/forum?id=On5WIN7xyD&noteId=trfqKNaK2D), and the download link can be found at the bottom of that page. We'd be happy to address any questions after you've reviewed the additional results and appreciate your willingness to reevaluate based on this information.

---

> > > > ### Comment · Reviewer_oXqB · 2024-08-13
> > > >
> > > > Thank you, I have looked at the pdf now and the results presented look satisfactory to me.

---

### Official Review · Reviewer_M7jM · 2024-07-12

**Soundness:** 3
**Presentation:** 3
**Contribution:** 3
**Rating:** 7
**Confidence:** 3

**Summary:**

This paper demonstrates the correlation, known as scaling laws, between training FLOPs and large language models’ (LLMs) downstream task abilities. The authors decompose performance metrics to fit this “Observational Scaling Law” and confirm its validity across emergent capabilities, agentic capabilities, and post-training interventions.

**Strengths:**

1. Sufficient validation. Including 21 model families and general, reasoning and agent-related datasets (open LLM  leadboard)
2. Clear formalization.  Observational Scaling Law in Eq.3, Eq.7.

**Weaknesses:**

I did not find any specific weaknesses in this paper.

**Questions:**

1. Why using PCA to decompose the main component ? This may need to explain why using PCA is work.   Do these main components have any corresponding physical significance?
2.  I believe this paper aims to predict the performance of future large models without the need for training, but it only considers FLOPs. In reality, data quality and model size should also be considered. Is there any analysis of these two factors?

**Limitations:**

See in Question.

---

> ### Author Rebuttal · Authors · 2024-08-06
>
> Thank you for your valuable feedback and suggestions. We appreciate that you acknowledged our paper provides sufficient validation and clear formalization. We would like to address your remaining questions and concerns in the following responses.
>
> ### Choice of using PCA
>
> > *”Why using PCA to decompose the main component ? This may need to explain why using PCA is work. Do these main components have any corresponding physical significance?”*
>
> Our motivating hypothesis is the existence of a low-rank space of capability measures that captures the major model benchmark performance, and PCA is the most standard method for low-rank decomposition. The PC components extract the most significant, orthogonal directions in the capability space that capture the different capability dimensions of LMs (as illustrated in Figure 2b), making it a desirable choice for our purpose. We have also conducted an additional analysis with nonnegative matrix factorization (included in Figure 2 in the supplementary PDF) and a detailed discussion on the tradeoff in [our response to Reviewer V1YP](https://openreview.net/forum?id=On5WIN7xyD&noteId=qBHgXOUGb4).
>
> ### Additional factors
>
> > *”I believe this paper aims to predict the performance of future large models without the need for training, but it only considers FLOPs. In reality, data quality and model size should also be considered. Is there any analysis of these two factors?”*
>
> We would like to clarify that our predictions are based on the low-dimensional capabilities measures extracted from the benchmarks – which project models from different training recipes (e.g., trained on different data) to a unified space. This enables us to utilize a lot of public models without needing to deal with their training heterogeneousness – which is one of the biggest advantages of our observational scaling approaches. In our results, our scaling law predictions have already accommodated models trained from different data (e.g., specifically trained on multilingual data like BLOOM, or those on code like StarCoder) or with different sizes (e.g., Llama-2 7B, 13B, 70B). The model-specific factors like the data quality are incorporated into the “compute/data efficiency” by which each model family converts FLOPs to capabilities, as seen by how different model families behave differently in their scaling properties on the capability dimensions (Figure 3 in the paper).
>
> We welcome any suggestions for additional critical analyses that you believe would further validate our findings, and we would be glad to conduct them for additional validation.

---

### Official Review · Reviewer_JzvB · 2024-07-12

**Soundness:** 3
**Presentation:** 3
**Contribution:** 3
**Rating:** 7
**Confidence:** 2

**Summary:**

The paper proposes a generalized class of scaling laws that encompass multiple model families of different sizes. These resulting scaling laws are capable of predicting “emergent” behaviors, complex agentic performance, and inference techniques in an extrapolative manner, as seen with GPT-4. The observed laws yield less error compared to standard FLOP-based laws.

**Strengths:**

1. The paper is well-written, with extensive experiments covering major benchmarks.
2. A unified scaling law for various families and benchmarks is important to the whole community.
3. The framework is general and demonstrates strong extrapolation performance.

**Weaknesses:**

See questions.

**Questions:**

1. How does the method apply to Llama 3, where the amount of data is significantly larger than the standard Chinchilla optimal? Can we still achieve accurate predictions for Llama 3 models?
2. For complex tasks, increasing amounts of compute are dedicated to inference time, especially for agentic benchmarks. Can the scaling law capture the scaling trends for inference time compute?
3. Are there any insights on how to choose the PC-dimension to balance preventing overfitting with achieving better estimation?

**Limitations:**

Limitations are discussed. The authors claim they do not foresee any direct societal impact from their work.

---

> ### Author Rebuttal · Authors · 2024-08-06
>
> Thank you for your valuable feedback and suggestions. We appreciate that you acknowledged that our work is “important to the whole community” with extensive experiments and strong extrapolation performance. We would like to address your remaining questions and concerns in the following response.
>
>
> ### Applicability to Llama-3
>
> > “*How does the method apply to Llama 3, where the amount of data is significantly larger than the standard Chinchilla optimal? Can we still achieve accurate predictions for Llama 3 models?*”
>
> We would like to clarify that our main results already included models that are significantly overtrained such as Llama-3 and Qwen, as shown in Figures 4 & 6. We have also further tested the newly released Llama-3.1 405B (FP8) with results included in Figure 1 in the supplementary PDF. We observe fairly accurate predictions on these models from observational scaling laws.
>
>
> ### Generalization to inference-time scaling
>
> > *”For complex tasks, increasing amounts of compute are dedicated to inference time, especially for agentic benchmarks. Can the scaling law capture the scaling trends for inference time compute?”*
>
> That is a good question! Our results on the agentic setups (with > 100 output tokens per step) and on the analysis of CoT (as another type of spending inference for capability) indicate the possibility of predicting model performance on inference-compute-intensive setups. Furthermore, we believe our observational scaling laws can be generalized to account for the inference time compute, e.g., by combining the capabilities measures of pretrained models and inference-time quantities (e.g., inference compute) to study the scaling properties of specific inference-time techniques with respect to certain model capabilities. This is an interesting direction, and we leave it for future work.
>
>
> ### PC selection
>
> > *”Are there any insights on how to choose the PC-dimension to balance preventing overfitting with achieving better estimation?”*
>
> We would like to note that we have conducted an extensive ablation study on the selection of the number of PCs in Appendix C.3. In our results, using three PCs generally leads to the best extrapolation performance across different setups. Furthermore, for a more fine-grained PC selection, we may utilize the total explained variances of the included PCs as a practical unsupervised selection criterion to balance representativeness and noise inclusion. We have also done some preliminary experiments, where we found selecting the smallest number of PCs (for preventing overfitting) with a total variance above 97% (for keeping representativeness) could select the best number in most of our setups.

---

### Official Review · Reviewer_V1YP · 2024-07-14

**Soundness:** 3
**Presentation:** 3
**Contribution:** 3
**Rating:** 7
**Confidence:** 4

**Summary:**

The paper proposes observational scaling laws, a method to use benchmark scores of LLMs to infer how their performance would change if the amount of training compute was scaled, without actually having to train additional models. The authors apply PCA to a model-task benchmark matrix in order to obtain latent capabilities, and find that only a small set of such capabilities can explain almost all of the benchmark variation. They use the latent capabilities to infer how efficient different model families are at converting compute into these capabilities, and to infer how models will perform on a new benchmark, based on their capabilities. The authors then examine how these observational scaling laws can be use to predict model performance on emergent, agentic and prompt-induced abilities and find that the predicted scaling performance closely matches the actual performance of held-out models.

**Strengths:**

1. The proposed observational scaling laws are a valuable contribution towards anticipating LLM abilities as they scale:
    - They are helpful in extrapolating various LLM abilities (emergent, agentinc, prompting-based, and possibly other abilities), which can help in clarifying some of the scaling debates around these abilities, e.g. such as for emergent abilities, and which gives practitioners a tool for anticipating the performance gains of different methods, such as of prompting strategies.
    - They do so purely based on observational benchmark performance, i.e. they alleviate the need for costly (interventional) training of additional models at different scales.
    - They seem to obtain reliable estimates by doing a joint inference over different model families and benchmarks.
2. The paper conducts a comprehensive analysis spanning benchmarks for 77 models from 21 families which demonstrate that the proposed observational scaling laws, using latent capability estimates, can predict the performance of more capable models from less capable ones, with low error and better than naive methods that directly estimate performance from compute or model size.
3. ​The paper offers some interesting insights:
    - The authors show that emergent LLM abilities may be explainable by a lack of "sampling resolution" in terms of number and performance of models, corroborating recent results in that area.
    - The latent capability model allows for inferring how much the performance on different benchmarks contributes to abilities (e.g. agentic and prompting-induced ones), showing that for instance performance on coding benchmarks strongly contributes to agentic and chain-of-thought abilities.
4. Overall, the paper is well written.

**Weaknesses:**

1. The conceptual model proposed in the paper is that the (principal component) PC-dimensions (latent capabilities) obtained based on model-benchmark performances capture basic, supposedly interpretable skills (e.g. general, reasoning, and programming capabilties) that increase the more training compute is applied to the models. However, as shown in Figure E.2 in the appendix, for some models for PC-2 and PC-3 there is a negative or flat relationship between training compute and model skill, as measured by the corresponding PC. Additionally, the performance on some of the benchmarks correlates negatively with some of the PCs, e.g. -0.62 for HellaSwag and PC-2 (which supposedly captures reasoning abilities). Given these discrepancies, I am not sure that the mental model of interpretable skills is really applicable for the PCs (even though they they seem to be useful for the scaling predictions).
2. In Section 4.2 and Figure 5 the authors show that observational scaling laws can be used to extrapolate the performance of LLMs in agent settings. However, the test sets used here are quite small (2 for AgentBench, 1 for AgentBoard), which makes these conclusions somewhat unreliable. Perhaps a different training-test split (e.g. 80/20 or 70/30) could be used (instead of 90/10) in order to obtain a larger test set. Would the results still hold?
3. There are some minor clarity issues
    1. Section 3.1 defines an error metric ($E_m$) which is somewhat unclear, i.e.
        - Is $E_m$ capturing errors or performance (~= 1 - errors)? Perplexity is mentioned as an example, but shouldn't perplexity decrease the more compute is applied to the model (Eq. 1)?
        - Is it always normalized? Line 111 mentions that, but I am not sure about the initial definition.
    2. I found it a bit difficult to understand Figures 5c and 6c from the captions. It would be useful to mention in the caption that they are based on $\beta^T \gamma$.

**Questions:**

1. Would there be benefits to using more fine-grained benchmarks to estimate the skill vector $S_m$, for example a detailed breakdown over different BigBench tasks?
2. Would it be possible to use the observational scaling laws to estimate the effect of finetuning? I.e. suppose we knew for a few models how finetuning them on a particular dataset changed their performance on some target measure. Could we apply the observational scaling laws to infer the performance of finetuning more capable models?
3. How many models per family would be needed in order for direct extrapolation from compute/model size for that family to produce similarly accurate results as the observational scaling laws here?
4. The conceptual model of latent capabilities that determine benchmark performance is strongly reminiscent of [item response theory](https://en.m.wikipedia.org/wiki/Item_response_theory) (IRT) from psychometrics, which underpins exams such as GRE. In IRT the underlying skills also determine a sigmoid-shaped probability curve, according to which a student will be able to answer a question with a particular difficulty correctly. It would be quite interesting to draw a connection between IRT and the latent capability model used by the observational scaling laws.

**Limitations:**

The paper does not explicitly discuss limitations. I believe that the discussion throughout the paper is sufficient, though there may still be some potential limitations that the authors do not touch upon, for intance:
1. How well do the scaling laws hold up if benchmark data has leaked into training?
2. All models within the same family are assumed to share the same compute-efficiency. I think that this is a valid assumption, because models in the same family are by and large trained with the same architecture and dataset, but it's conceivable that there may be more heterogeneous families.

---

> ### Author Rebuttal · Authors · 2024-08-06
>
> Thank you for your valuable feedback and suggestions. We appreciate that you acknowledged our paper offers a “valuable contribution” with a comprehensive analysis and interesting insights. We would like to address your remaining questions and concerns in the following response.
>
>
>
> ### Interpretability of the capability dimensions
>
> > *”However, as shown in Figure E.2 in the appendix, for some models for PC-2 and PC-3 there is a negative or flat relationship between training compute and model skill, as measured by the corresponding PC. Additionally, the performance on some of the benchmarks correlates negatively with some of the PCs, e.g. -0.62 for HellaSwag and PC-2 (which supposedly captures reasoning abilities). Given these discrepancies, I am not sure that the mental model of interpretable skills is really applicable for the PCs (even though they they seem to be useful for the scaling predictions).”*
>
> This is a really good point!
>
> First, we would like to note that the negative scaling of PC-2 & PC-3 for some model families is due to their negative coefficients on certain benchmarks for decorrelation with PC-1. We agree that this may undermine the interpretability of these PCs to some extent.
>
> Second, we would like to note that interpretability is *not a must* for the conceptual foundation of observational scaling laws, especially for making the scaling predictions that we mostly care about (as the reviewer has mentioned). What we need is the existence of low-rank capability measures that robustly correlate with compute measures and downstream capabilities and enable the scaling predictions from small to large scales, which have been validated in the paper.
>
> Finally, if interpretability is really needed in the extracted capability measures, we can apply [non-negative matrix factorization](https://en.wikipedia.org/wiki/Non-negative_matrix_factorization) which enforces the non-negativity of coefficients. We have conducted the analysis and included the results in Figure 2 in the supplementary PDF:
> - In Figure 2(a), we visualize the component-wise benchmark coefficients. We observe that the coefficients are enforced to be non-negative and provide an interpretable decomposition. For example, we may view Component-1 as reasoning (that emphasizes GSM8K, HumanEval, and MMLU) and Component 4 as language understanding capabilities (that emphasizes (X)Winograde, HellaSwag), respectively.
> - In Figure 2(b), we visualize the scaling of NMF components with respect to log FLOPs for each family, using Component 1 as an example. We observe a smooth, positive scaling that generally holds across model families, which also hold for other components (omitted due to space constraints).
>
> While NMF offers enhanced interpretability and positive scaling properties compared to PCA, it also has notable limitations. Firstly, unlike PCA, NMF does not enforce orthogonality among its extracted components, as evident in the observed correlation between Components 3 and 4. Consequently, the coefficients assigned to each model across dimensions may not serve as independent measures of specific capabilities. Secondly, the ordering of NMF components lacks uniqueness and intrinsic physical meaning. This contrasts with PCA components, which are systematically ordered by their explained variances. The PCA approach provides an 'importance' measure for each dimension and allows for controlled trade-offs between representativeness and noise inclusion by adjusting the number of PCs used in the analysis.
>
> We will adjust the claims and include the discussion in our future version of the paper.
>
>
> ### Number of test points in agentic setups
>
> > *”However, the test sets used here are quite small (2 for AgentBench, 1 for AgentBoard), which makes these conclusions somewhat unreliable. Perhaps a different training-test split (e.g. 80/20 or 70/30) could be used (instead of 90/10) in order to obtain a larger test set. Would the results still hold?”*
>
> We would like to clarify that the small number of test points is due to the small number of models evaluated by AgentBench and AgentBoard, and that a sufficient number of data points need to be used in the training set for fitting reliable scaling laws. We look forward to testing our current scaling predictions on more capable models once they are released, similar to what we have done in other setups (Figure 1(a) in the supplementary PDF). We will also adjust the claims about these results in the paper to acknowledge the limited number of test points.
>
> We have also tested the 80/20 train/test split on AgentBench where there are more available data points. The results are included in Figure 1(b) in the supplementary PDF. We observe although the extrapolation performance tends to underestimate the performance to some extent, it still aligns with the overall observed trend.
>
>
>
> ### Scaling extrapolation within a single model family
>
> > *“How many models per family would be needed in order for direct extrapolation from compute/model size for that family to produce similarly accurate results as the observational scaling laws here?”*
>
> This is an interesting question! It is hard to do a comprehensive apple-to-apple comparison since most accessible model families contain only a few data points (e.g., Llama-2) or without public compute measures (e.g., GPT).  We did preliminary experiments with the Qwen1.5 family (with 7 models) and OPT (with 8 models), and tested them on tasks where the scaling predictions are non-trivial. The results are included in Figure 3 in the supplementary PDF. We find that typically at least 5 models are required for accurate extrapolation but the performance is highly dependent on the speciﬁc setup. For example, using five Qwen1.5 models achieves decent extrapolation on word unscramble but poor extrapolation on Persian QA.

---

> ### Author Response · Authors · 2024-08-06
> **Additional Response**
>
> ### Clarity
>
> > *”Is $𝐸_𝑚$ capturing errors or performance (~= 1 - errors)? Perplexity is mentioned as an example, but shouldn't perplexity decrease the more compute is applied to the model (Eq. 1)?”*
>
> In our formulation, $𝐸_𝑚$ refers to the model error (~= 1 - accuracy). For the case of perplexity, it does decrease with more compute, which corresponds to the case where $\beta_f < 0$ (we did not constrain $\beta_f$ to be nonnegative). Due to the potential confusion it may cause, we will adjust the formulation with a negative sign included for $\beta_f$.
>
>
> > *”Is it always normalized? Line 111 mentions that, but I am not sure about the initial definition.”*
>
> It does not have to be normalized in the general case (e.g., perplexity). For our specific case in Eq 2 & 6 with the use of a sigmoid nonlinearity for downstream predictions, normalization is required.
>
> > *”I found it a bit difficult to understand Figures 5c and 6c from the captions. It would be useful to mention in the caption that they are based on $\beta^{T}\gamma$”*
>
>
> Thanks for your suggestions. We will update the captions with the suggested clarifications in our future version of the paper.
>
>
> ### Other questions & suggestions
>
> > *”Would there be benefits to using more fine-grained benchmarks to estimate the skill vector 𝑆𝑚, for example a detailed breakdown over different BigBench tasks?”*
>
> Fine-grained benchmarks could help in cases where the available benchmarks may not be able to form a capability space that sufficiently captures the dynamics of downstream capabilities. For example, if the downstream task is related to "a scientific research agent", then the more fine-grained performance on the "Professional" split of MMLU could potentially offer additional predictive gains.
>
> In our case, the included benchmarks seem to well capture the downstream tasks that we have tested.
>
>
>
> > *”Would it be possible to use the observational scaling laws to estimate the effect of finetuning?... Could we apply the observational scaling laws to infer the performance of finetuning more capable models?”*
>
> This a good idea! We believe it is possible to establish scaling laws for fine-tuning, by combining the capabilities measures of pretrained models and finetuning-related quantities (e.g., fine-tuning data size) to study the scaling properties of specific finetuning techniques with respect to certain model capabilities. This is an interesting direction, and we leave it for future work.
>
>
>
> > *”The conceptual model of latent capabilities that determine benchmark performance is strongly reminiscent of item response theory (IRT) from psychometrics,”*
>
> Thanks for the suggestions! We will include a discussion of the connection to IRT in our future version of the paper.
>
>
>
> ### Limitation discussion
>
> > *”I believe that the discussion throughout the paper is sufficient, though there may still be some potential limitations that the authors do not touch upon, for instance, …”*
>
> Thanks for the suggested limitation discussion! We acknowledge that they are indeed parts of the limitations of our work and will more extensively discuss them in a separate section in the future version of the paper.

---

> ### Comment · Reviewer_V1YP · 2024-08-13
>
> I thank the authors for their thorough response to my questions!
>
> The additional results on NMF capabilities are interesting and this method might be helpful in settings where interpretability of the capability dimensions is needed. However, I think that it is easy to read too much meaning into the capabilities, and it is important to make sure that any interpretations of the capabilities are consistent with their actual task-relationship. The authors and I seem to be in agreement that ultimately the most important function of the capabilities is to provide accurate estimates of benchmark performances, which they seem to be achieving.
>
> It is great to see the additional results with the 80/20 split on AgentBench, as well as the addition of the Llama-3.1-405B model.
>
> I also think the comparison with non-observational scaling laws Qwen and OPT models shows that the observational approach seems to be able to more accurately forecast the performance of scaled models, while requiring less models per family.
>
> I maintain my positive score and support the acceptance of the paper.

---

### Author Rebuttal · Authors · 2024-08-06

We thank all reviewers for their helpful feedback and suggestions.

We are glad that the reviewers found our work offers a valuable contribution [V1YP] and very promising progress [oXqB] toward an important problem [oXqB, JzvB] with a comprehensive analysis [V1YP], extensive experiments [JzvB, M7jM], and interesting insights [V1YP].

We have provided extensive empirical results (included in the supplementary PDF) and responses to address reviewers’ remaining questions. Specifically, we have attempted to address the following major ones:
- [Reviewer oXqB, DfgG, V1YPm] Pushing back the cutoff points on emergent capability [oXqB, DfgG] and agentic tasks [V1YPm, DfgG]: In individual responses, we have clarified the reason for the use of our current cutoff selection. Furthermore, we have also done additional analysis on pushing back the cutoff selection on each setup for robustness check and tested the newly released **Llama-3.1-405B** for scalability check, included in Figure 1 in the PDF.
- [Reviewer V1YP, M7jM] The use of PCA for capability decomposition and its interpretability: We have analyzed an additional low-rank decomposition method (non-matrix factorization) in Figure 2 in the PDF and discussed its tradeoff against PCA in individual responses.
- [Reviewer V1YPm] Scaling extrapolation within a single model family: We have analyzed the scaling prediction from FLOPs within a single family and included results in Figure 3 in the PDF.
- [Reviewer oXqB] Confidence intervals: We have calculated the 95% confidence intervals for predictions, included in Figure 4 in the PDF.

Please see detailed responses to specific questions below.

---

### Decision · Program_Chairs · 2024-09-25

**Decision:**

Accept (spotlight)

**Comment:**

The paper introduces Observational Scaling Laws, a method to use benchmark scores of LLMs to infer how their performance would change if the amount of training compute was scaled, without actually having to train additional models. The proposed method applies PCA to the benchmark performances and finds that a small set of PCs (3) can explain much of the benchmark variation. The results show that the latent PCs can infer how models will perform on a new benchmark and how efficient models are at using their compute.

All 5 reviewers and I are in agreement that the contributions are important and timely, and are supported by extensive experiments on 77 models and 21 model families. The additional results presented on the newly released Llama3.1 model during the rebuttal also strengthens the contribution and shows that this work has promise to predict capabilities of new models.

The authors were responsive to the reviewer's few concerns in their rebuttal and I urge the authors to update their paper with the promised changes, especially the addition of confidence intervals on the predictions and a separate section detailing the possible limitations of the work.